# Synapsin 2a tetramerisation selectively controls the presynaptic nanoscale organisation of reserve synaptic vesicles

Shanley F. Longfield[1], Rachel S. Gormal [1], Matis Feller [2], Pierre Parutto [2], Jürgen Reingruber[2], Tristan P. Wallis [1], Merja Joensuu [1,6], George J. Augustine [3], Ramón Martínez-Mármol [1], David Holcman[2,4] & Frédéric A. Meunier [1,5] ✉

Neurotransmitter release relies on the regulated fusion of synaptic vesicles (SVs) that are tightly packed within the presynaptic bouton of neurons. The mechanism by which SVs are clustered at the presynapse, while preserving their ability to dynamically recycle to support neuronal communication, remains unknown. Synapsin 2a (Syn2a) tetramerization has been suggested as a potential clustering mechanism. Here, we used Dual-pulse sub-diffractional Tracking of Internalised Molecules (DsdTIM) to simultaneously track single SVs from the recycling and the reserve pools, in live hippocampal neurons. The reserve pool displays a lower presynaptic mobility compared to the recycling pool and is also present in the axons. Triple knockout of Synapsin 1-3 genes (SynTKO) increased the mobility of reserve pool SVs. Re-expression of wild-type Syn2a (Syn2a^WT), but not the tetramerization-deficient mutant K337Q (Syn2a^K337Q), fully rescued these effects. Single-particle tracking revealed that Syn2a^K337QmEos3.1 exhibited altered activity-dependent presynaptic translocation and nanoclustering. Therefore, Syn2a tetramerization controls its own presynaptic nanoclustering and thereby contributes to the dynamic immobilisation of the SV reserve pool.

Neuronal communication relies on the fusion of synaptic vesicles (SVs) with the plasma membrane of the presynaptic bouton, and the release of neurotransmitters contained within SVs into the synaptic cleft. Due to their small size (45 nm), SVs, like small molecules, are subjected to thermal energy which tends to randomize their distribution within the 3D volume of the neuron. Despite such randomizing forces, electron microscopy studies have demonstrated that SVs are organized in clusters within the presynapse[1,2], via a mechanism that is not fully understood. SVs can be categorized into three major presynaptic pools defined by their release probabilities: a readily releasable pool, a recycling pool and a reserve pool[3]. The readily releasable pool consists of SVs that are typically docked, primed and released immediately upon arrival of a presynaptic action potential[4]. The recycling pool is comprised of any SV that recycles upon moderate stimulation (typically 10–20% of all SVs)[5,6]. This process requires SV fusion with the plasma membrane, and subsequent SV reformation from the plasma membrane via compensatory endocytosis. Newly formed SVs are then refilled with neurotransmitters for another round of fusion. Shortly

¹Clem Jones Centre for Ageing Dementia Research, Queensland Brain Institute, The University of Queensland, Brisbane QLD 4072, Australia. ²Group of Data Modelling and Computational Biology, IBENS, Ecole Normale Superieure, 75005 Paris, France. ³Temasek Lifesciences Laboratory, Singapore, Singapore. ⁴Department of Applied Mathematics and Theoretical Physics (DAMPT) visitor, University of Cambridge, and Churchill College, CB30DS Cambridge, UK. ⁵School of Biomedical Sciences, The University of Queensland, Brisbane QLD 4072, Australia. ⁶Present address: Australian Institute for Bioengineering and Nanotechnology, The University of Queensland, Brisbane QLD 4072, Australia. ✉e-mail: f.meunier@uq.edu.au

after endocytosis, recycling SVs can be found throughout the nerve terminal closely intermixed with the SV cluster[3] (the SV cluster is typically defined as an accumulation of SVs found adjacent to the active zone). The pool of recycling SVs can therefore remain connected to the SV cluster while still displaying the high mobility required to translocate to the plasma membrane, undergo fusion, and subsequent recycling from the plasma membrane back to the SV cluster[3,7]. The reserve pool contains the majority of SVs, the contents of which are only released in response to high-frequency stimulation and depletion of the recycling pool[3,8–10]. Lastly, SVs found within the nearby axonal segment have been classified as a superpool and have been implicated in synaptic plasticity[11].

The fact that SVs are highly concentrated in presynaptic terminals suggests that one or more anchoring mechanism(s) exist. One potential anchor are Synapsins–a family of highly conserved phosphoproteins that interact with SVs in the presynaptic bouton[12–16]. In humans, Synapsin proteins are encoded by 3 genes (SYN1, SYN2, and SYN3)[17–20], and have been shown to bind to the phospholipids of SVs via their N-terminus or to other SV proteins via their proline-rich C-terminus[21,22]. Synapsins have been proposed to act as gatekeepers, allowing SVs from the reserve pool to replenish the recycling pool of SVs under high-frequency stimulation[2,14,15,18,23]. Subsequently, Synapsin-dependent interactions are thought to regulate several aspects of neuronal communication; alterations in Synapsin expression levels also result in epileptic phenotypes[2]. The mechanism by which Synapsins control the anchoring of SVs is still debated. A recent discovery that Synapsin molecules can form phase-separated droplets or condensates via liquid-liquid phase separation (LLPS) has led to the proposal that such droplets can trap SVs[24]. Evidence supporting this hypothesis includes observations that Synapsin condensates are capable of clustering small liposomes[24] and SVs[25] in vitro. Further, injection of antibodies raised against the intrinsically disordered region (IDR) of Synapsin dramatically reduces the number of SVs clustered at the lamprey presynapse[26], and blockade of Synapsin LLPS via the SH3A domain of intersectin impairs SV clustering and GABA release at inhibitory synapses[25].

An alternative hypothesis has been proposed that relies on the ability of Syn2a to oligomerize and concomitantly bind to SVs[21,27,28] to mediate their clustering. Dimerization[28,29] and tetramerization[30] of Synapsins have been proposed to cross-link SVs, thereby generating a connected network[2,16,28,31]. A point mutation in Syn2a, which was found to disrupt a hydrogen bond critically involved in its tetramerization (K337Q)[30,32], dramatically impacts its ability to cluster SVs in vitro and impairs the mobilization of SVs from the reserve pool of excitatory synapses[25].

One of the main obstacles in assessing SV clustering has been the static nature of current experimental observations: most studies have focused on electron microscopy and fluorescence microscopy of fixed neurons. More recently, super-resolution microscopy techniques have been developed to track single molecules in their native environment[33,34] within live neurons and have revealed critical changes in the dynamic nanoscale organization of molecules implicated in vesicular priming[35,36]. Further, a quantitative technique called sub-diffractional Tracking of Internalized Molecules (sdTIM) has been developed as a way to track individual recycling SVs in live hippocampal neurons[37,38]. Ideally, individual SVs from both the recycling and reserve pools would be tracked simultaneously to identify any specific roles Synapsins play in the clustering and mobility of SVs. Furthermore, the simultaneous tracking of both pools would allow for the quantitative assessment of their targeting of individual nerve terminals and their ability to translocate to and from the nearby superpool (axonal population of SVs)[11].

Here, we have met this challenge by developing a super-resolution single-particle tracking protocol named Dual-pulse sdTIM (DsdTIM) to monitor the spatiotemporal dynamics of both recycling and reserve SVs simultaneously. As expected, we observed recycling pool SVs in both the presynaptic and axonal compartments of hippocampal neurons. Surprisingly, reserve pool SVs were also observed in the axonal compartment, suggesting a more dynamic organization than previously imagined. Both pools were found to be enriched at presynaptic terminals as compared to neighbouring axonal segments. DsdTIM imaging in hippocampal neurons from Synapsin triple knockout (SynTKO) mice showed an increase in the mobility of reserve pool SVs. This increase in mobility was rescued upon expression of wild-type Synapsin 2a (Syn2a$^{WT}$). However, expression of the tetramerization-deficient Synapsin 2a mutant (Syn2a$^{K337Q}$) only partially rescued this effect, demonstrating that Syn2a tetramerization controls its own presynaptic nanoclustering and thereby contributes to the dynamic immobilization of the SV reserve pool.

## Results

### Recycling synaptic vesicles become more immobile over time suggesting a transition into the reserve pool

Tracking single SVs in the crowded environment of nerve terminals in living neurons is a challenge that can be overcome by various super-resolution imaging techniques. One such technique, sdTIM, is a single-particle tracking super-resolution localization microscopy approach that can characterize the dynamics of the recycling pool of SVs in live neurons[37,38]. Over time, recycling SVs lose their ability to undergo fusion as they become part of the reserve pool of SVs[6,39]. In this study, we took advantage of this time-dependent transition to image and track individual SVs and quantify their mobility over an extended time frame (10 min to 72 h) as they transitioned from the recycling pool to the reserve pool (Fig. 1a). Synaptotagmin1 (Syt1) is a vesicular protein that is transiently associated with the plasma membrane following exocytic fusion and is then re-internalized into recycling SVs[40]. We imaged single SVs using a modified sdTIM protocol[37,38], which allowed us to label both recycling and reserve SVs. Briefly, hippocampal neurons grown on glass-bottom dishes were transfected at 14 days in vitro (DIV14) with a Syt1 construct tagged with pHluorin (Syt1pH). Neurons were then treated with a medium containing high K$^+$ to trigger exocytosis, and anti-green fluorescent protein (GFP) Atto565–tagged nanobodies (At565Nb) to label the pHluorin tag of Syt1pH (pulse). The neurons were then washed to remove unbound At565Nb and incubated for various periods of time in conditioning medium (chase), to characterize the transition of SVs from the recycling pool to the reserve pool. Time-lapse (50 Hz) super-resolution imaging of the presynaptic and axonal compartments of live neurons allowed us to visualize and track single Syt1pH-bound At565Nb that had been internalized into SVs (Fig. 1b-e). With this approach, we could precisely track and quantify the movement of individual SVs[41]. Single-particle tracking data were analyzed to derive the mean square displacement (MSD) of Syt1pH/At565Nb-tagged SV trajectories. We performed statistical comparisons using the area under the MSD curve (AUC). The data were further processed using custom spatiotemporal analysis software[42] to obtain a map representation of the instantaneous diffusion coefficient of the Syt1pH/At565Nb trajectories showing SV mobilities at 10 min (Fig. 1c) and 48 h (Fig. 1e) after the pulse. These images qualitatively show the lower mobility of Syt1pH/At565Nb trajectories at 48 h, indicating that SVs had undergone the transition from recycling to reserve pool at this time point. This compliments previous work[39] which demonstrated that SVs labelled 48 hours prior to imaging no longer co-localized with newly labelled SV proteins and were no longer fusogenic, suggesting that these 'old' vesicles segregated from the recycling pool into the reserve pool. Further quantification of all time points showed a progressive decrease in SV mobility over time, specifically within the presynaptic terminal (Fig. 1f, g), but not in the axonal compartment (Fig. 1h, i). The lowest SV mobility in the presynaptic compartment was observed at 48 h, confirming that by this time recycling SVs had transitioned into the reserve pool.

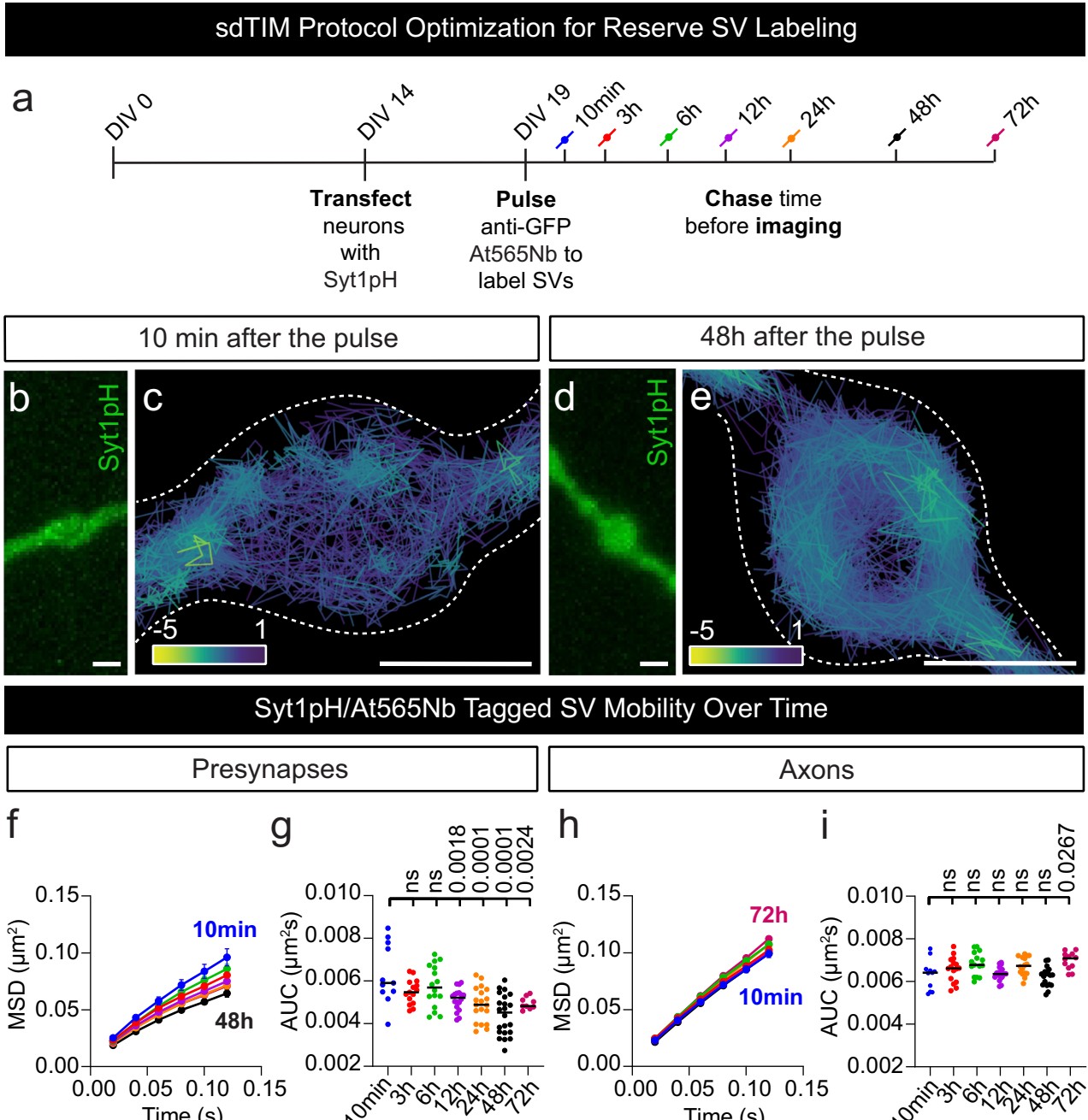

**Fig. 1 | Tracking the reserve pool of SVs in live hippocampal neurons.**
**a** Graphical representation of the subdiffractional Tracking of Internalized Molecules (sdTIM) protocol optimization for the tracking of reserve pool synaptic vesicles (SVs): Days in vitro (DIV) 19 hippocampal neurons expressing Synaptotagmin1–pHluorin (Syt1pH) were stimulated with high K$^+$ medium containing anti-green fluorescent protein (GFP) Atto565-tagged nanobodies (At565Nb; red) for one minute. Following stimulation, the excess nanobodies were washed off, and the neurons were chased for various time periods in conditioning medium as indicated in the figure, and then imaged in a low K$^+$ imaging buffer. Representative (**b**, **d**) epifluorescence Syt1pH images and (**c**, **e**) corresponding Syt1pH/At565Nb SV trajectory maps (colour coded by their instantaneous diffusion coefficients; colour bar represents Log$_{10}$ [μm$^2$s$^{-1}$]) of presynapses chased for (**b**, **c**) 10 min and (**d**, **e**) 48 h. **f**, **h** Average mean square displacement (MSD) of the trajectories generated from Syt1pH/At565Nb trajectories in (**f**) presynapses and (**h**) axons at different time points. **g**, **i** Area under the MSD curve (AUC; μm$^2$s) in (**g**) presynapses and (**i**) axons. Data are displayed as mean ± SEM. Values in (**f**–**i**) were obtained from $n = 11$ presynapses and $n = 11$ axons (10 min chase), $n = 15$ presynapses and $n = 16$ axons (3 h chase), $n = 16$ presynapses and $n = 15$ axons (6 h chase), $n = 18$ presynapses and $n = 18$ axons (12 h chase), $n = 19$ presynapses and $n = 16$ axons (24 h chase), $n = 22$ presynapses and $n = 19$ axons (48 h chase) and $n = 9$ presynapses and $n = 12$ axons (72 h chase) from ≥3 independent neuronal cultures. Statistical comparisons were performed using the one-way ANOVA and Dunnett's or Tukey's multiple comparisons test in (**g**) and (**i**). The different chase time points in (**f**–**i**) were compared to the 10 min chase time point, which was considered a recycling SV mobility control. Scale bars 1 μm (**b**–**e**). Source data are provided as a Source Data file.

## Reserve and recycling SVs display distinct mobility patterns at the presynapse

We next attempted to simultaneously image individual SVs from both the recycling and the reserve pools of SVs. For this purpose, we expanded upon the modified sdTIM protocol (single pulse), to develop a technique that incorporates a second pulse (Dual-pulse sdTIM; DsdTIM). In brief, DsdTIM uses two spectrally-distinct anti-GFP nanobodies to combine the labelling approach for reserve SVs (labelled 48 h

prior to imaging) with that of recycling SVs (labelled 10 min prior to imaging) (Fig. 2a). Specifically, a high K$^+$ pulse with At565Nb was performed followed by a 48 h chase to allow for the labelled SVs to mature into reserve pool SVs. The reserve SVs were then imaged using single-channel excitation (561 nm) to capture their resting state mobility (Syt1pH/At565Nb; resting reserve SVs). Immediately following this, a second high K$^+$ pulse was applied, along with spectrally distinct anti-GFP Atto647N nanobodies (At647Nb) to label nascent recycling SVs (Syt1pH/At647Nb; recycling SVs). Following this second pulse and a subsequent 10 min chase, dual-channel excitation (561 nm and 647 nm) was performed to enable simultaneous imaging of both SV pools in both the presynaptic and axonal compartments. Syt1pH fluorescence was used to generate regions of interest for nerve terminals and neighbouring axonal segments (Fig. 2b) in order to analyze SV trajectories in these two areas (Fig. 2c–h). In addition to labelling different pools of SVs in the presynaptic compartment, DsdTIM has the ability to track SVs within the axonal compartments, thus capturing SVs transiting between synapses (superpool[11]). We detected the presence of both recycling and reserve SVs in the axonal compartment, suggesting that the superpool is not solely comprised of recycling SVs as previously reported[11] (specifically defined as recently endocytosed SVs[7]). Spatiotemporal analysis of these data highlights that both pools reside within the presynapses and axons, and that SVs are less mobile within the nerve terminals, indicating that the presynaptic environment confines SVs (Supplementary Fig. 1). The reserve pool is defined as the population of SVs released only upon intense stimulation such as strong, prolonged high K$^+$ depolarization[3]. We therefore imaged the reserve pool immediately before (resting reserve pool (RP)) and after (stimulated reserve pool (RP)) the second pulse of high K$^+$ to confirm that these SVs did not respond to this short high K$^+$ stimulation protocol. We quantified the mobility of the reserve (resting RP and stimulated RP) and recycling SV pools within the presynaptic and axonal compartments using the MSD (statistically compared using AUC) (Fig. 3a–d) and average diffusion coefficients (Fig. 3e, f) of the Syt1pH/At565Nb (reserve SV) and Syt1pH/At647Nb (recycling SV) trajectories. As shown in Fig. 3a, b, the mobility of SVs at 48 h (resting RP) in the presynapse was not affected by the second pulse (stimulated RP), indicating that this pool is indeed insensitive to high K$^+$ stimulation as expected. This validates the ability of DsdTIM to distinguish between the reserve and recycling pools of SVs. The mobility of the recycling pool was significantly higher than that of the resting and stimulated reserve pools in both the presynapse and nearby axonal segments (Fig. 3a–d). We further assessed these data by extracting the average diffusion coefficient of the reserve (resting RP and stimulated RP) and recycling pool of SVs in both the presynapses and axons (Fig. 3e, f), which showed the same mobility trends as the MSD and AUC data: reserve and recycling pools displayed significantly different mobilities, with the mobility of the reserve pool remaining unaffected by stimulation. Furthermore, we quantified the density of reserve and recycling SV trajectories in presynapses and axons and found that for both pools the density was higher in presynapses (Fig. 3g, h). This indicates that SVs cluster in boutons and that the superpool harbours SVs from both pools. To confirm that the difference in mobility between the reserve and recycling pool of SVs was not due to the use of the two different anti-GFP tags (At565Nb and At647Nb), we performed a single pulse with both tags added simultaneously. Under these conditions, there was no significant difference in the mobilities of SVs tagged with either At565Nb or At647Nb at either the 10 min or 48 h time point (Supplementary Fig. 2), validating that the mobility of probed SVs depends solely upon the chase time. Further, to validate that the nanobody is selectively internalized into GFP-positive SVs, we applied anti-GFP-At647Nbs to hippocampal neurons transfected with cytosolic TagBFP, which is not extracellular facing, and could not detect any internalized/bound nanobodies (Supplementary Fig. 3). Lastly, to test whether the depolarizing

stimulus (high K$^+$) captured the dynamics of the reserve pool of SVs, we tested a more physiological stimulation paradigm. We performed high-frequency field stimulation (50 Hz, 300 action potentials (APs)) in sterile conditions in mature hippocampal neurons expressing Syt1pH (as above) in the presence of the At647Nb and chased for 48 h. The mobility of the labelled vesicles was indistinguishable from those observed following high K$^+$ stimulation (Supplementary Fig. 4), thereby validating our high K$^+$ stimulation protocol. To further characterize the differences between the recycling and reserve pools of SVs, we used a high-throughput statistical approach that constructs diffusion and density maps of trajectories to identify regions of both high density and low diffusion[43–45]. The underlying biophysical model (Eq. 2 in Methods) assumes that SVs can either move according to Brownian motion or interact with the local environment to stabilize themselves within sub-micrometer subregions (See Supplementary Fig. 5). The density and diffusion maps were constructed from the ensemble of trajectories by estimating the local density and diffusion coefficients in a grid map decomposed into small bins (see Methods for the statistical estimators equations 5–9). The results are shown in Fig. 4a–j. The density map reveals areas (red) characterized by local high-density regions (HDRs) for recycling and reserve SV populations of live hippocampal neurons (Fig. 4g–j illustrates examples of potential wells or SV traps in these pools). The diffusion map reveals a more uniform distribution of SV mobility within the axonal compartment (Fig. 4e, f). To analyze these HDRs, we used the potential paradigm framework[43] under which such regions result from long-range force interactions. We first observed that HDRs are much more frequent for reserve ($n = 77$) than recycling SVs ($n = 4$) for the same neuronal region. We further characterized these HDRs by extracting their boundaries and associated energy using an automated classification algorithm[46]. We found that the size of the wells associated with the reserve SV pool was larger ($0.18 \pm 0.07\,\mu m$) than those for the recycling SV pool ($0.1 \pm 0.01\,\mu m$) (Supplementary Fig. 5). In addition, the stability of the reserve pool measured by the energy of the well was higher ($E = 2.15$ $kT$) than the stability of the recycling pool ($E = 0.94\,kT$) (Fig. 4k). Lastly, we determined the residence time in confinement versus Brownian diffusion for trajectories within the wells. We found that the potential wells were able to confine reserve SVs for at least 3x times longer than SVs of the recycling pool (Fig. 4l). The reserve pool of SVs therefore has a much larger number of HDRs defined as potential wells. These wells are larger in size (Supplementary Fig. 5) and their energy is also larger compared to that of the recycling pool (Fig. 4k). This (1) confirms that the two populations of DsdTIM-labelled SVs are indeed different (2) suggests that the reserve pool is much more stable, and (3) the associated mechanism of SV trapping involves a greater force than that of the recycling pool.

## SynTKO affects reserve SV mobility at the presynapse

Having established a technique to observe the reserve pool of SVs, we next sought to investigate the role of SV-associated protein Synapsin in mediating the dynamics of this SV pool. Synapsin, which exists as 3 isoforms (SYN1-3), has long been suggested as a molecular tether that holds SVs in a connected cluster distal from the synaptic active zone[25,31]. To examine the role of Synapsins, we used hippocampal neurons from Synapsin triple knockout (SynTKO) mice[47] (Fig. 5). Following the expression of Syt1pH in the SynTKO hippocampal neurons, we used DsdTIM to determine the mobility of both reserve and recycling SV pools in nerve terminals and neighbouring axonal segments (Fig. 5a–e). Surprisingly, the complete absence of Synapsin did not significantly alter the mobility patterns of either the reserve or recycling pools in boutons (Fig. 5f, g) or axons (Fig. 5h, i). The reserve pool continued to be less mobile than the recycling pool, and its mobility still was unaffected by high K$^+$ stimulation. However, comparing results from SynTKO and wild-type (WT) neurons revealed that the mobility of both pools was higher in the SynTKO neurons (Fig. 6a, b).

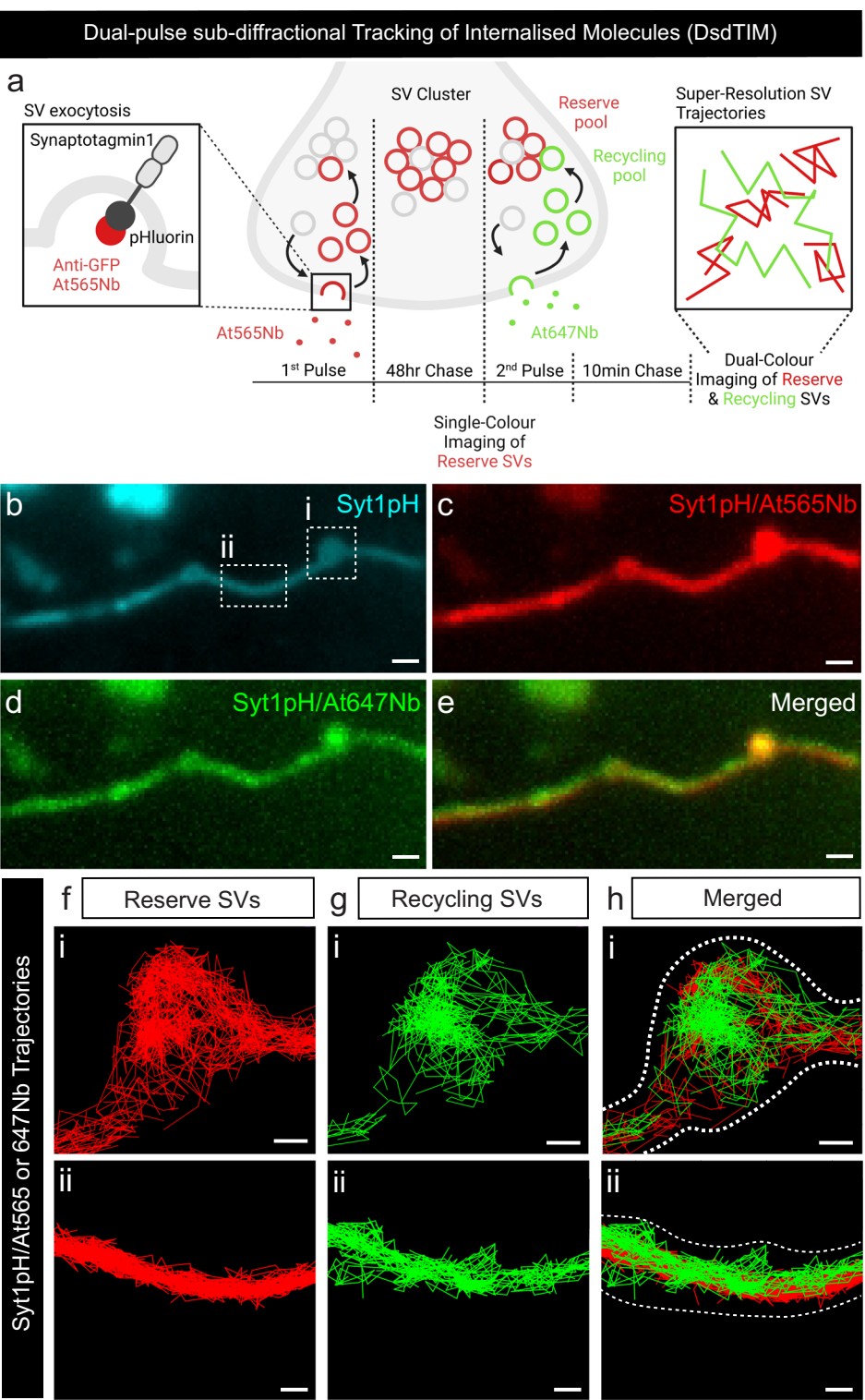

**Fig. 2 | Simultaneous tracking of the reserve and recycling pool of SVs in live hippocampal neurons using DsdTIM. a** Graphical representation of the Dual-pulse sub-diffractional Tracking of Internalized Molecules (DsdTIM) protocol. Hippocampal neurons expressing Synaptotagmin1-pHluorin (Syt1pH) were stimulated for one min with high $K^+$ medium containing anti-green fluorescent protein (GFP) Atto565-tagged nanobodies (At565Nb; red). After stimulation, the excess nanobodies were washed off, and the neurons were chased for 48 h in conditioning medium. After the 48 h chase, the labelled SVs were imaged to assess their mobility at 50 Hz. Immediately afterwards, the same neurons were stimulated for a second time, pulsing for five min with high $K^+$ imaging buffer containing anti-GFP Atto647N-tagged nanobodies (At647Nb; green). After this re-stimulation, the excess At647Nb were washed off, and the neurons were chased for 10 min in a low $K^+$ imaging buffer. To

detect nanobodies inside individual SVs, the neurons were imaged using dual-colour single-molecule tracking at 50 Hz. Graphic created with BioRender.com. **b** Representative epifluorescence image of an axonal segment expressing Syt1pH acquired before incubation with At647Nb. The dashed boxes in (**b**) highlight (**i**) a presynaptic compartment and (**ii**) a peri-synaptic axonal segment. **c–e** Maximum intensity projection of (**c**) Syt1pH/At565Nb (reserve pool; 48 h chase), (**d**) Syt1pH/At647Nb (recycling pool; 10 min chase), and (**e**) merged maximum intensity projection. **f** Trajectory map of tracked reserve SVs containing Syt1pH bound by At565Nb in the (**i**) presynapse and (**ii**) axonal segment. **g** Trajectory map of tracked recycling SVs containing Syt1pH bound by At647Nb in the (**i**) presynapse and (**ii**) axonal segment. **h** Merged trajectory maps. Scale bar 1 μm (**b**–**e**) and 200 nm (**f**–**h**). Data were obtained from ≥3 independent neuronal cultures.

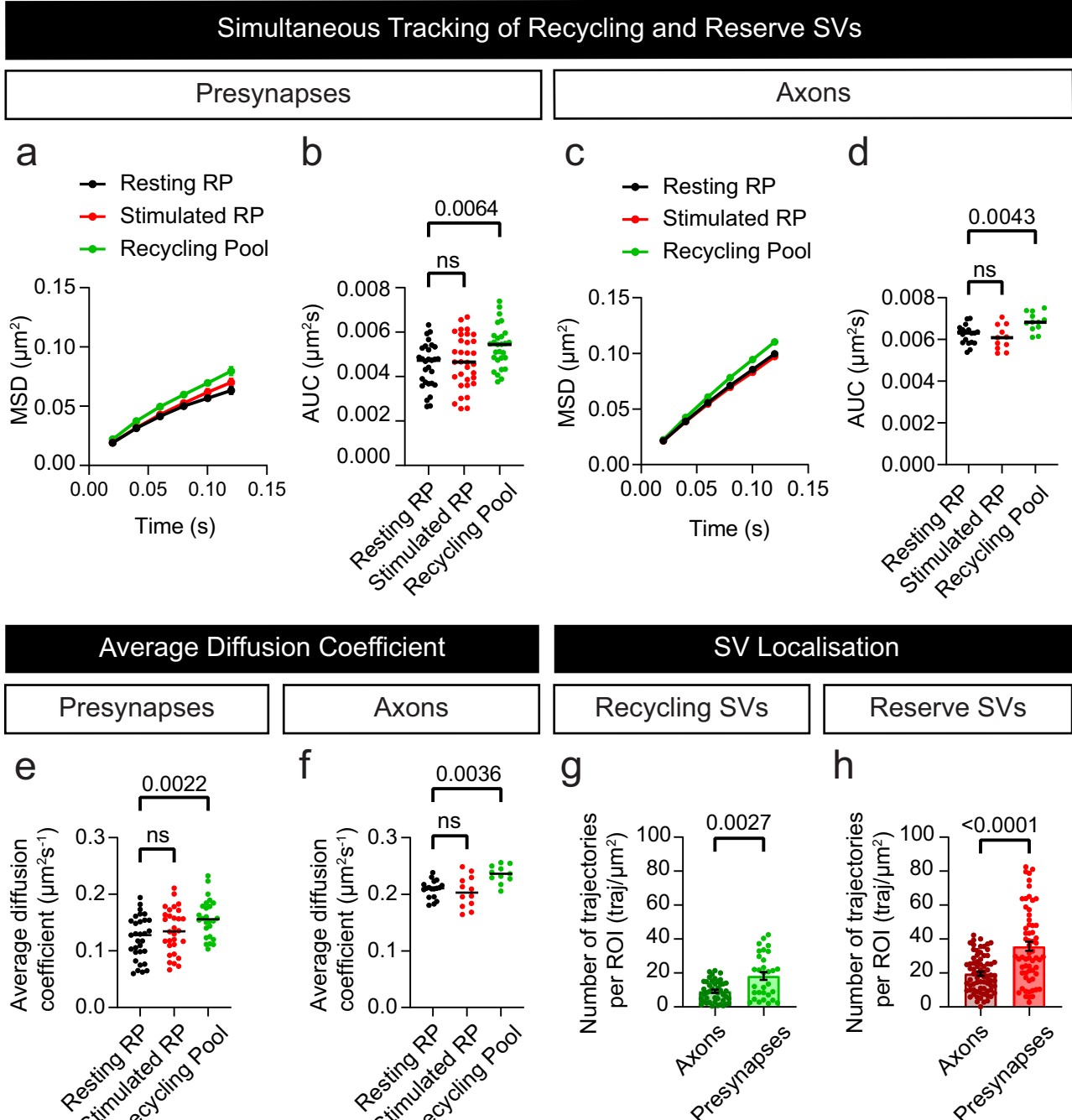

**Fig. 3 | Quantification of the reserve and recycling SV pool mobilities.**
**a**, **c** Average mean square displacement (MSD; μm²) of the resting reserve pool (RP) of synaptic vesicles (SVs; black), RP SVs after stimulation (red) and the recycling pool of SVs (green) within the (**a**) presynapses and (**c**) axons. **b**, **d** Area under the MSD curve (AUC; μm²s) for (**b**) presynapses and (**d**) axons. **e**, **f** Average diffusion coefficient of the resting RP (black), RP SVs after stimulation (red) and the recycling pool of SVs (green) within the (**e**) presynapses and (**f**) axons. **g**, **h** Comparison of the density of detections in the axons and presynapses from (**g**) recycling and (**h**) reserve SVs labelled with anti-green fluorescent protein (GFP) Atto565 or 647N-tagged nanobodies (At565Nb or At647Nb) respectively, normalized by the area of the region of interest (ROI; traj/μm²). Data are displayed as mean ± SEM. Values were obtained from $n$ = 29 presynapses (Resting RP), $n$ = 33 presynapses (Stimulated RP) and $n$ = 28 presynapses (Recycling Pool) in (**a**, **b**), from $n$ = 19 axons (Resting RP), $n$ = 11 axons (Stimulated RP) and $n$ = 12 axons (Recycling Pool) in (**c**, **d**), from $n$ = 31 presynapses (Resting RP), $n$ = 30 presynapses (Stimulated RP) and $n$ = 26 presynapses (Recycling Pool) in (**e**), from $n$ = 18 axons (Resting RP), $n$ = 12 axons (Stimulated RP) and $n$ = 11 axons (Recycling Pool) in (**f**), from $n$ = 31 presynapses and $n$ = 45 axons in (**g**) and $n$ = 61 presynapses and $n$ = 74 axons in (**h**). Data were obtained from ≥3 independent neuronal cultures. Statistical comparisons were performed using the one-way ANOVA and Dunnett's or Tukey's multiple comparisons test in (**b**, **d**–**f**) and the unpaired two-tailed Mann–Whitney $U$ test in (**g**) and (**h**). Source data are provided as a Source Data file.

This effect was specific to presynaptic terminals and was not observed in axons (Fig. 6c, d). Thus, Synapsins selectively lower the mobility of SVs within presynaptic terminals. Analysis of the density of detections in axons versus boutons shows that the recycling and reserve pools are preferentially targeted to presynaptic terminals (Fig. 6e, f). When looking at the ratio of recycling to reserve SVs in WT and SynTKO neurons, we observed a 3-fold increase in the number of recycling SVs in the SynTKO neurons indicating a potential shift in the sorting/

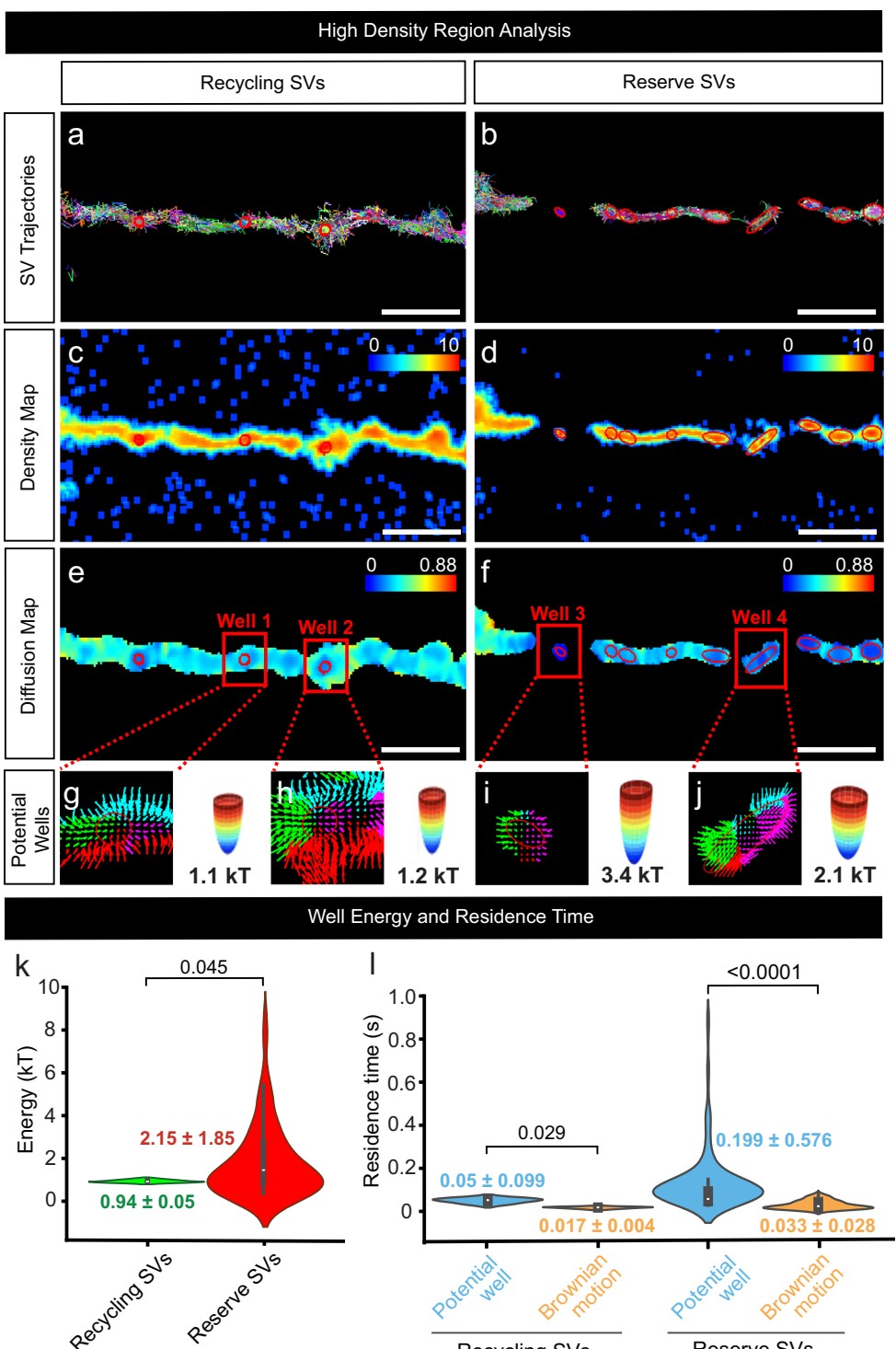

**Fig. 4 | Density and diffusion maps describing the differences between reserve and recycling SVs. a**, **b** Representative neuron where (**a**) recycling and (**b**) reserve synaptic vesicle (SV) trajectories are displayed using different colours. Red ellipses show potential wells (see **g**–**j**). **c**, **d** Density map of (**c**) recycling and (**d**) reserve SVs. The unit corresponds to the logarithm of the number of trajectories per bin. **e**, **f** Diffusion maps of (**e**) recycling and (**f**) reserve SVs in units of μm²/s. **g**–**j** Examples of potential wells of (**g**, **h**) recycling and (**i**, **j**) reserve SVs. Potential wells are high-density regions with converging drift vectors (4 colours corresponding to 4 main directions). The ellipses delimit the boundaries of the potential wells. The energy of the wells is given in units of kT. **k** Violin plots showing the energy distribution of the potential wells for recycling SVs (green) and reserve SVs (red). The energy mean ± SD are displayed in the figure. The white bars show the median (0.92 kT for recycling and 1.45 kT for reserve), and the black bars indicate the interquartile ranges Q1, Q2, Q3 and Q4. The minimal and maximal energies are 0.89 and 1.03 kT for recycling

wells, and 0.37 and 9.47 kT for reserve wells. **l** Violin plots showing the resident time in the potential wells (blue) and the resident time expected with simple Brownian motion (orange) for recycling SVs (left) and reserve SVs (right). The mean ± SD are displayed in the figure. The white bars show the median 0.052 s and 0.019 s for recycling SVs, and 0.060 s and 0.024 s for reserve SVs. The black bars indicate the interquartile ranges Q1, Q2, Q3 and Q4 (Q1 values: 0.045 s and 0.017 s for recycling SVs and 0.046 s and 0.014 s for reserve SVs; Q3 values: 0.057 s and 0.019 s for recycling SVs, and 0.10 s and 0.045 s for reserve SVs; minimal times: 0.034 s and 0.011 s for recycling SVs, and 0.028 s and 0.0040 s for reserve SVs; maximal times: 0.061 s and 0.021 s for recycling SVs, and 3.73 s and 0.211 s for reserve SVs). The data in (**k**, **l**) were obtained from ≥3 independent neuronal cultures. The distributions in (**k**, **l**) were computed from n = 4 recycling wells and n = 77 reserve wells. The statistical test used to compute the p-values is a two-sample Kolmogorov–Smirnov test for goodness of fit. Source data are provided as a Source Data file.

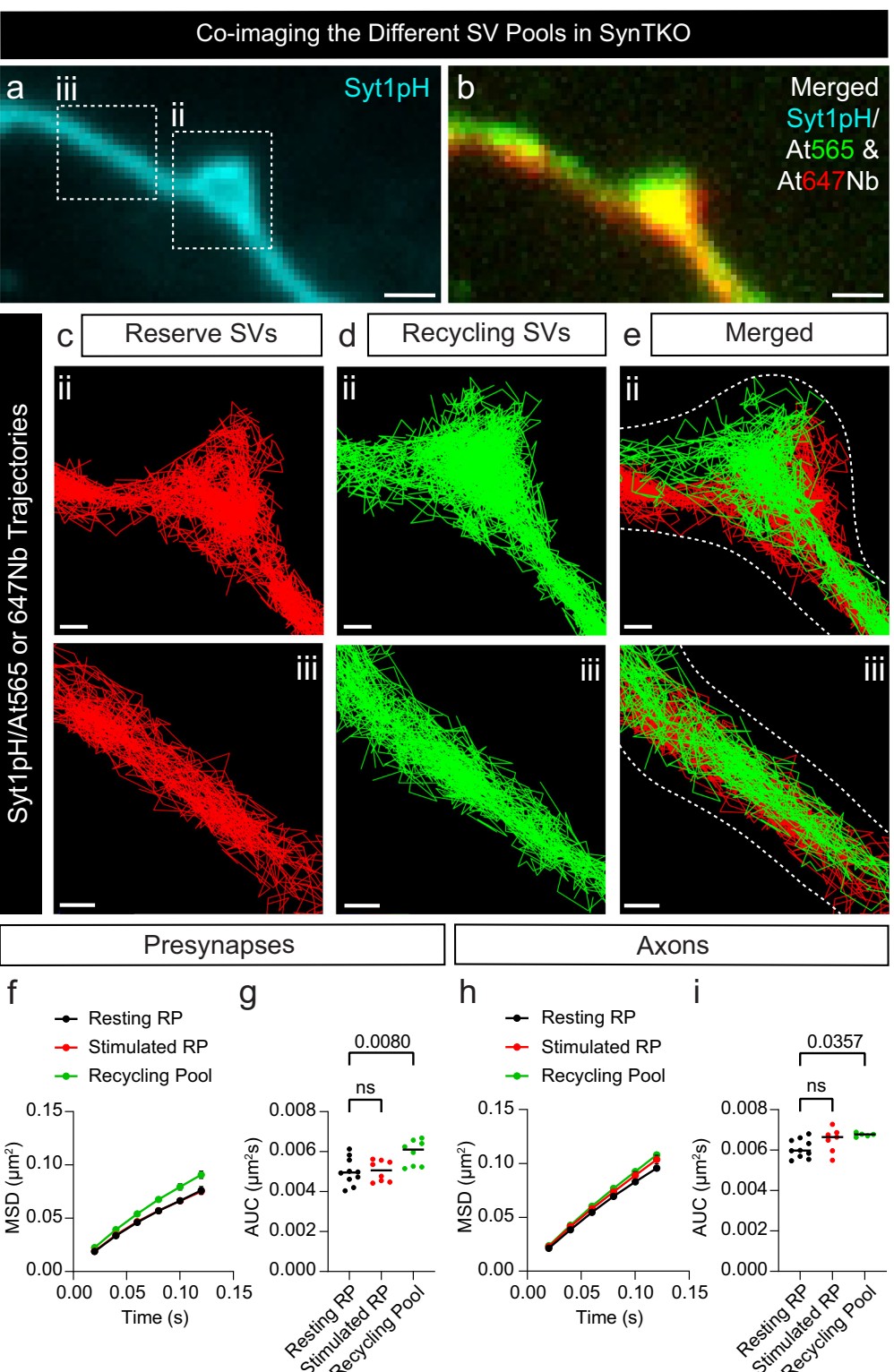

**Fig. 5 | Simultaneous tracking of the reserve and recycling pool of SVs in live SynTKO hippocampal neurons. a** Representative epifluorescence image of a neuronal segment of a Synapsin triple knockout (SynTKO) hippocampal neuron, transfected with Synaptotagmin1-pHluorin (Syt1pH) acquired before addition of anti-green fluorescent protein (GFP) Atto565-tagged nanobodies (At565Nb). The dashed boxes in panel 'a' highlight a (**ii**) presynaptic compartment and (**iii**) peri-synaptic axonal segment. **b** Merged maximum intensity projection of Syt1pH-bound anti-GFP Atto647N-tagged nanobodies (Syt1pH/At647Nb; reserve pool; 48 h chase) and Syt1pH/At565Nb (recycling pool; 10 min chase). **c** Trajectory map of tracked reserve SVs containing Syt1pH/At565Nb in the (**ii**) presynapse and (**iii**) axonal segment. **d** Trajectory map of tracked recycling SVs containing Syt1pH/

At647Nb in the (**ii**) presynapse and (**iii**) axonal segment. (**e**) Merged trajectory maps. **f, h** Average mean square displacement (MSD; μm²) of resting reserve SVs (black), reserve SVs after stimulation (red) and recycling SVs (green) within the (**f**) presynapses and (**h**) axons. **g, i** Area under the MSD curve (AUC; μm²s) for (**g**) presynapses and (**i**) axons. Data are displayed as mean ± SEM. Values were obtained from $n$ = 24 presynapses (Resting RP), $n$ = 15 presynapses (Stimulated RP) and $n$ = 16 presynapses (Recycling Pool) in (**f, g**), from $n$ = 10 axons (Resting RP), $n$ = 7 axons (Stimulated RP) and $n$ = 5 axons (Recycling Pool) in (**h, i**). Data were obtained from three biological replicates. Statistical comparisons were performed using the one-way ANOVA and Tukey's multiple comparisons test in (**g**) and (**i**). Scale bar 1 μm (**a, b**) and 200 nm (**c–e**). Source data are provided as a Source Data file.

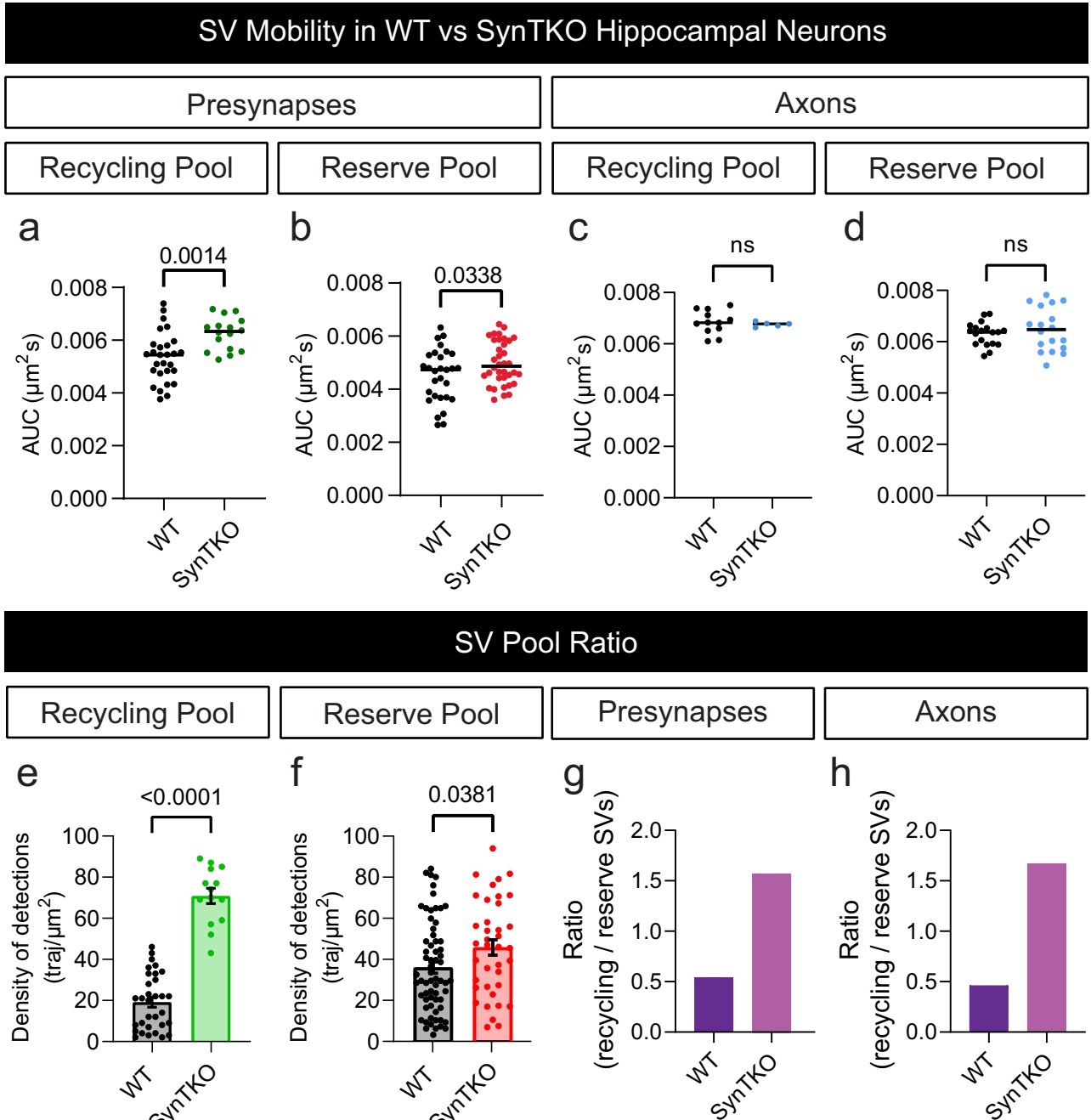

**Fig. 6 | Synapsin TKO selectively affects reserve vesicle mobility. a, b** Area under the average mean square displacement (MSD) curve (AUC; μm²s) of presynaptic (**a**) recycling and (**b**) reserve synaptic vesicles (SVs) from wild-type (WT) and Synapsin triple knockout (SynTKO) hippocampal neurons. **c, d** AUC (μm²s) of axonal (**c**) recycling and (**d**) reserve SVs from WT and SynTKO hippocampal neurons. **e, f** Comparisons of the density of detections from WT or SynTKO neurons from (**e**) recycling and (**f**) reserve SVs, normalized by the area (traj/μm²). **g, h** Ratio of recycling to reserve SVs in (**g**) presynapses and (**h**) axons for WT (dark purple) and SynTKO (light purple) hippocampal neurons. Data are displayed as mean ± SEM.

Values were obtained from $n$ = 28 WT presynapses and $n$ = 16 SynTKO presynapses in (**a**); from $n$ = 29 WT presynapses and $n$ = 36 SynTKO presynapses in (**b**); from $n$ = 12 WT axons and $n$ = 5 SynTKO axons in (**c**); from $n$ = 19 WT axons and $n$ = 18 SynTKO axons in (**d**); from $n$ = 32 WT presynapses and $n$ = 14 SynTKO presynapses in (**e**); from $n$ = 65 WT presynapses and $n$ = 38 SynTKO presynapses in (**f**). Ratios in (**g**, **h**) were obtained from the values in (**e**, **f**). Data were obtained from three biological replicates. Statistical comparisons were performed using the unpaired two-tailed Student's $t$ test in (**a**–**d**) and the unpaired two-tailed Mann–Whitney $U$ test in (**e**) and (**f**). Source data are provided as a Source Data file.

segregation between the two populations of SVs (Fig. 6g, h). The altered ratio of recycling to reserve SVs in the SynTKO neurons could reflect a loss of gatekeeping within the presynaptic boutons. It has been reported that approximately 80% of SVs are in the reserve pool in WT neurons[3]. In agreement with this, we observed a higher proportion of reserve SVs to recycling SVs in wild-type neurons. In SynTKO neurons, the relative proportion of recycling to reserve SVs was lower. This

is consistent with electron microscopy images showing a selective loss of reserve pool SVs in SynTKO presynaptic terminals[24,47]. Our data further suggests that Synapsin functions as a gatekeeper that participates in the transition/segregation between the two SV pools (Fig. 6e–h).

The mechanism by which Synapsin mediates this effect is currently up for debate. Previous reports indicate that Syn2a is the only

isoform capable of rescuing the loss of SV clustering observed in central synapses of SynTKO neurons[48]. It has been suggested that Syn2a forms tetramers that link SVs together[2,25,30]. To investigate the role of Syn2a tetramerization in SV clustering, we first asked whether tetramerization affected the mobility of Syn2a, and then determined whether tetramerization of Syn2a impacts reserve SV pool mobility. To do this, we generated mEos3.1-tagged wild-type Synapsin 2a (Syn2a$^{WT}$-mEos3.1) and tetramerization-deficient mutant (Syn2a$^{K337Q}$-mEos3.1) constructs, which were then expressed in SynTKO hippocampal neurons. We performed single-particle tracking photoactivated localization microscopy (sptPALM) on unstimulated and stimulated neurons, which allowed us to quantify the mobility and localization of Syn2a-mEos3.1 molecules (Fig. 7). Trajectories within representative SynTKO presynaptic hippocampal nerve terminals were pseudo-coloured to qualitatively illustrate the instantaneous diffusion coefficients of individual Syn2a molecules (Fig. 7a, b). These images reveal a higher mobility of the Syn2a$^{K337Q}$-mEos3.1 mutant compared to Syn2a$^{WT}$-mEos3.1. Quantitative analyses of Syn2a mobility (Fig. 7c–f) show that the tetramerization-deficient mutant consistently displayed higher mobility than the wild-type. This effect was less pronounced in axons (Fig. 7e, f). These data indicate that tetramerization contributes to Syn2a immobilization within presynaptic terminals.

### Synaptic activity controls Synapsin 2a nanoscale organization through its ability to tetramerise

Next, we examined the effect of high K$^+$ stimulation on the mobility of Syn2a expressed in SynTKO neurons, with Syt1pH co-expressed as a marker of presynaptic boutons. Analyses of the mobility of Syn2a$^{WT}$-mEos3.1 showed an activity-dependent increase in its mobility as reflected by an increase in the MSD and AUC (Fig. 8a, b). In contrast, this effect was reduced in the tetramerization-deficient Syn2a mutant, which was already highly mobile (Fig. 8c, d). This indicates that synaptic activity regulates the nanoscale organization of Syn2a. We also observed that stimulation significantly decreased the number of trajectories detected for both wild-type Syn2a and the tetramerization-deficient Syn2a mutant in the axons (Fig. 8e, g), and concomitantly increased the detections of trajectories of the wild-type, but not the mutant, in presynaptic terminals (Fig. 8f, h). These results suggest that Syn2a$^{WT}$ can diffuse from axons to presynaptic boutons in an activity-dependent manner. Importantly, this effect depends on the ability of Synapsins to tetramerize (Fig. 8g, h).

### Synapsin 2a re-expression rescues reserve SV mobility in SynTKO neurons

Finally, we assessed whether Syn2a could rescue the altered mobility of the reserve pool found in SynTKO neurons (see Fig. 5). SynTKO neurons were transfected with Syt1pH and either Syn2a$^{WT}$-mEos3.1 or Syn2a$^{K337Q}$-mEos3.1, pulsed with At647Nb for 1 min in high K$^+$ neurobasal medium on DIV18, then washed and chased for 48 h to label the reserve pool (Fig. 9a). Dual-colour imaging allowed the visualization of both the reserve pool of SVs (Syt1pH-At647Nb) and individual Syn2a molecules (Syn2a$^{WT}$-mEos3.1 or Syn2a$^{K337Q}$-mEos3.1). Merging of all fluorescent channels showed clear co-localization of WT Syn2a with the reserve SVs within the presynaptic terminal (Fig. 9b–e). Analysis of the mobility of the reserve pool revealed that WT Syn2a expression rescued the high mobility phenotype of reserve pool SVs in the presynapses of SynTKO neurons (Fig. 9f, g). Importantly, the tetramerization-defective Syn2a$^{K337Q}$ mutant was unable to rescue this phenotype (Fig. 9f–i). Further, re-expression of Syn2a$^{WT}$-mEos3.1 in SynTKO neurons dramatically reduced the mobility of reserve SVs in presynapses, an effect that was significantly less pronounced for Syn2a$^{K337Q}$-mEos3.1 (Fig. 9j). Interestingly this effect was inverted in the axons with rescuing expression of Syn2a$^{WT}$-mEos3.1 increasing the mobility of the superpool of SVs, which was not achieved following re-

expression of Syn2a$^{K337Q}$-mEos3.1 (Fig. 9k). Following the re-expression of Syn2a$^{WT}$-mEos3.1 in SynTKO neurons, we observed an increase in the ratio of reserve SVs found in the presynaptic compartment compared to the axon, which was not observed following the re-expression of Syn2a$^{K337Q}$-mEos3.1 (Fig. 9l). Overall, our data indicates that tetramerization of Syn2a is required for selectively and dynamically anchoring SVs within the reserve pool of presynaptic terminals.

## Discussion

In this study we investigated the nanoscale dynamics of two major SV pools, the recycling and reserve, using advanced super-resolution imaging techniques and high throughput single-particle trajectory data analysis. Fluorescence Recovery After Photobleaching[49,50], Fluorescence Correlation Spectroscopy[51–53], time-lapse sparse labelling using styryl dyes[54,55] and Stimulated Emission Depletion microscopy have been pivotal in providing key information on isolated SVs and endosomal trafficking in live hippocampal terminals and axons[56]. While these methods have been instrumental in defining broad characteristics of the SV pools, they do not directly isolate and study the large and enigmatic 'reserve' pool of SVs. We overcame this limitation through implementing two temporally separated SV labelling pulses and optimizing our methodological approach to capture the incorporation of recycling SVs into the reserve pool. This has opened the way to simultaneously label and track the reserve and recycling pools of SVs within live hippocampal neurons and to characterize their distinct nanoscale organizations at both the presynapse and nearby axonal segments.

The reserve SVs are dynamically, albeit strongly, tethered to the presynapse when compared to recycling SVs. Both pools, however, exhibited a significant axonal pool suggesting that both pools are capable of exchanging vesicles between neighbouring nerve terminals. Although unexpected, this finding suggests, in addition to the recycling SVs, exchanging SVs from the reserve pool might also be important for synaptic plasticity[11]. In-depth analysis of single SV trajectories reveals that the most striking difference between the reserve and recycling pools stemmed from the number and size of high-density regions, which were much more prominent for the reserve pool of SVs. These high-density regions were both larger and more stable for the reserve pool, suggesting that the two types of vesicles interact differently with their nanoscale presynaptic environment. Surprisingly, high-density regions were also found along the axon which could indicate that these are silent synapses[57] and/or that the reserve pool has an intrinsic ability to generate clusters in axons. It remains unclear as to how these high-density regions are generated, and what mechanism(s) determine their size (which can extend for up to two hundred nanometres).

Furthermore, we established the role of Syn2a tetramerization in dynamically anchoring the reserve pool of SVs at the presynapse. Ultrastructural analysis of SynTKO neuronal synapses revealed dispersed SVs and altered SV clusters adjacent to the active zone[24]. Synapsins have been shown to undergo LLPS and mediate membraneless compartments called biomolecular condensates (BMCs) at the presynapses[24]. Presynaptic BMCs were recently shown to be dynamically regulated by phosphorylation[58], and tau phosphorylation in particular, was found to mediate the dynamic clustering of the recycling pool of SVs[58]. Although it is still unclear whether Synapsin forms BMCs that restrict SVs at the presynapse or form part of potential SV wells, the large intrinsically disordered regions (IDRs) of Synapsin 1 could play a role in generating BMCs[59]. In our study, synaptic activity dramatically increased the mobility of Syn2a$^{WT}$ suggesting that Synapsin becomes more diffusible in response to stimulation. This had a knock-on effect on the distribution of Syn2a$^{WT}$, with the number of detections decreasing in axons and increasing in presynaptic terminals. Similarly, it has been demonstrated that Tau also undergoes a switch in localization from the axon to the nearby presynapses where it forms nanoclusters likely to be nanoscale BMCs[58]. As both Tau and Synapsin

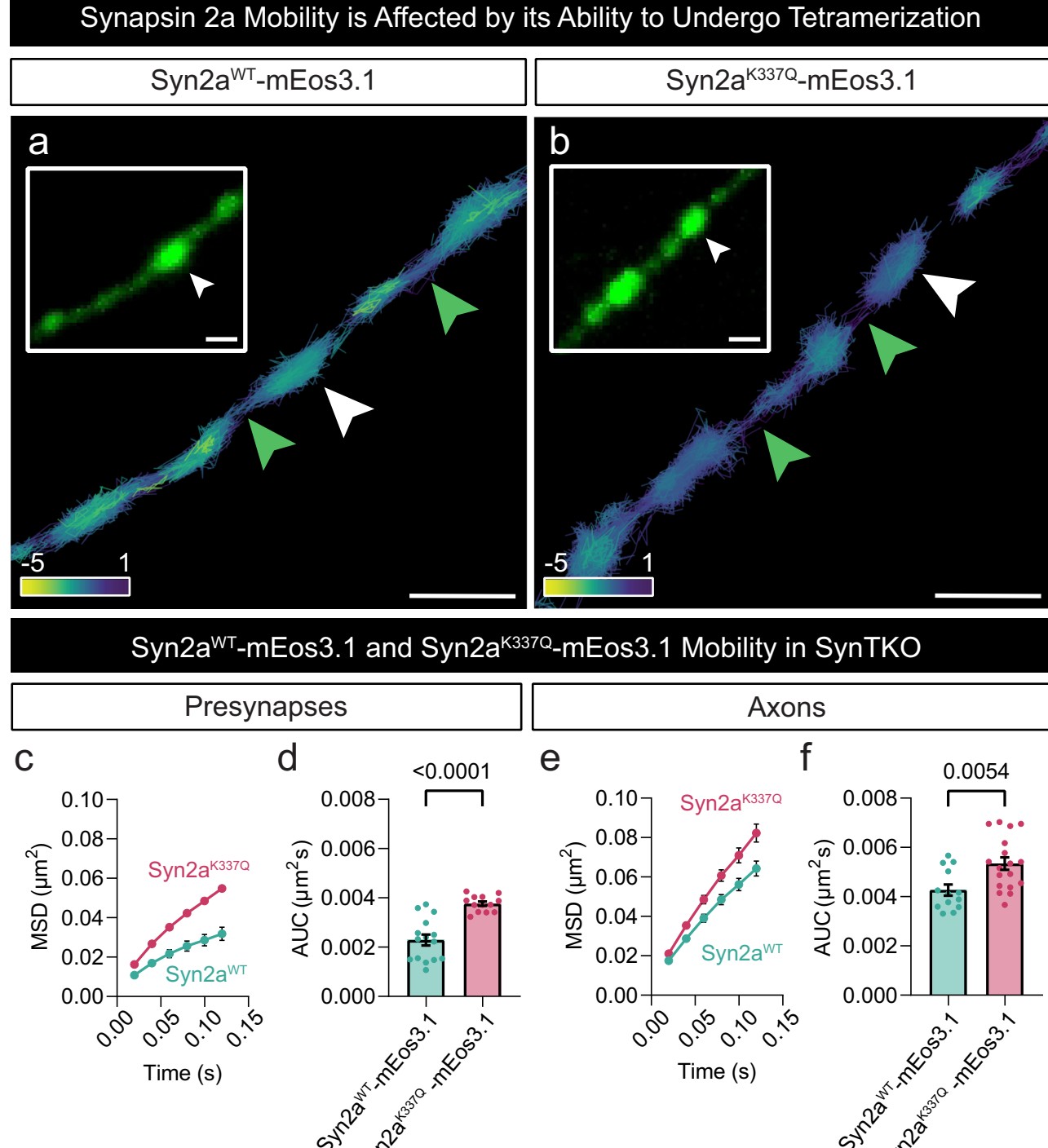

**Fig. 7 | Synapsin 2a mobility is sensitive to the protein's ability to tetramerize.**
**a**, **b** Representative trajectory maps (colour coded by their instantaneous diffusion coefficients; colour bar represents $Log_{10}$ [$\mu m^2 s^{-1}$]) of mEos3.1-tagged (**a**) wild-type (WT) Synapsin 2a (Syn2a$^{WT}$-mEos3.1) and (**b**) tetramerization deficient Synapsin 2a K337Q (Syn2a$^{K337Q}$-mEos3.1) in Synapsin triple knockout (SynTKO) hippocampal neurons. The insets are low-resolution TIRF images of the respective unstimulated presynaptic (white arrow) and axonal segments (green arrows) prior to photoconversion of mEos3.1, shown at a higher magnification. **c**, **e** Average mean square displacement

(MSD; $\mu m^2$) of Syn2a$^{WT}$-mEos3.1 (cyan) and Syn2a$^{K337Q}$-mEos3.1 (magenta) within (**c**) presynapses and (**e**) axons. **d**, **f** Area under the MSD curve (AUC; $\mu m^2 s$) for (**d**) presynapses and (**f**) axons. Data are displayed as mean ± SEM. Values were obtained from $n = 15$ Syn2a$^{WT}$-mEos3.1 transfected presynapses and $n = 13$ Syn2a$^{K337Q}$-mEos3.1 transfected presynapses in (**c**, **d**) and from $n = 13$ Syn2a$^{WT}$-mEos3.1 transfected axons and $n = 18$ Syn2a$^{K337Q}$-mEos3.1 transfected axons in (**e**, **f**). Data were obtained from 3 biological replicates. Statistical comparisons were performed using the unpaired two-tailed Student's $t$ test in (**d**) and (**f**). Source data are provided as a Source Data file.

regulate the nanoscale organization of the recycling and reserve pools respectively, it would be interesting to explore further the regulation of these two proteins at the presynapse by synaptic activity and presumably phosphorylation.

In addition to their ability to generate BMCs, both the N- and C-terminal IDRs of Synapsin can bind to lipid membranes, the actin-cytoskeleton, and synaptic proteins including other Synapsin isoforms[17]. A number of studies have used peptides of the E-domain

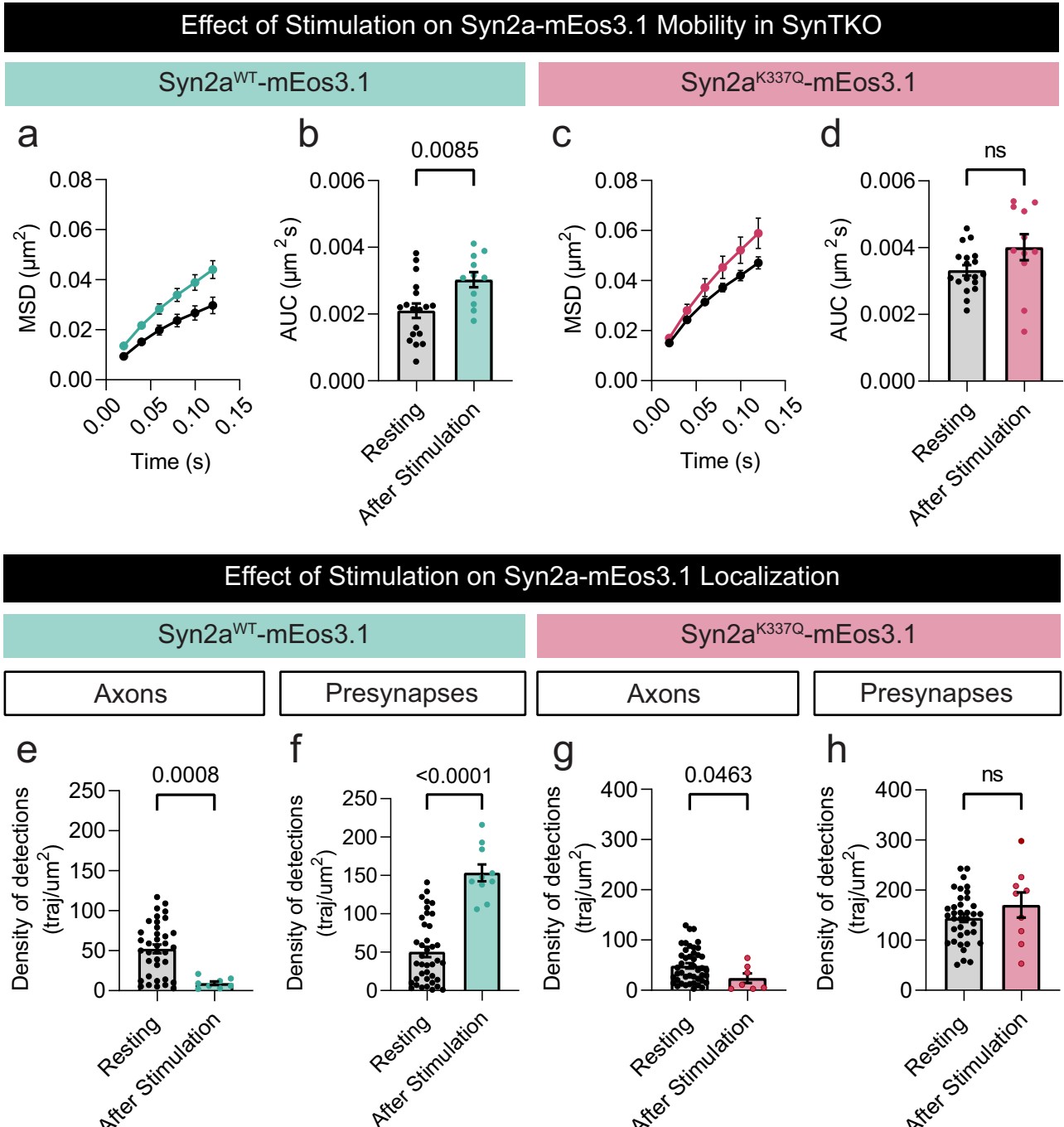

**Fig. 8 | Synapsin 2a's nanoscale organization is regulated by synaptic activity and this effect is tetramerisation-dependent. a, c** Average mean square displacement (MSD; $\mu m^2$) of mEos3.1-tagged (**a**) wild-type (WT) Synapsin 2a (Syn2a$^{WT}$-mEos3.1; cyan) and (**c**) tetramerization deficient Synapsin 2a K337Q (Syn2a$^{K337Q}$-mEos3.1) (magenta) within the presynapses before (resting) and after stimulation. **b, d** Area under the MSD curve (AUC; $\mu m^2$ s) of (**b**) Syn2a$^{WT}$-mEos3.1 (cyan) and (**d**) Syn2a$^{K337Q}$-mEos3.1 (magenta) within the presynapse, before (resting) and after stimulation. **e, f** The effect of stimulation on the number of Syn2a$^{WT}$-mEos3.1 trajectories detected in the (**e**) axonal and (**f**) presynaptic compartment of Synapsin triple knockout (SynTKO) hippocampal neurons normalized by the area (traj/$\mu m^2$). **g, h** The effect of stimulation on the number of Syn2a$^{K337Q}$-mEos3.1 trajectories

detected in the (**g**) axonal and (**h**) presynaptic compartments of SynTKO hippocampal neurons normalized by the area (traj/$\mu m^2$). Data are displayed as mean ± SEM. Values were obtained from $n = 18$ presynapses (Resting) and $n = 11$ presynapses (After Stimulation) in (**a–d**); from $n = 37$ axons (Resting) and $n = 8$ axons (After Stimulation) in (**e**); from $n = 39$ presynapses (Resting) and $n = 10$ presynapses (After Stimulation) in (**f**); from $n = 44$ axons (Resting) and $n = 7$ axons (After Stimulation) in (**g**) and from $n = 37$ presynapses (Resting) and $n = 9$ presynapses (After Stimulation) in (**h**). Data were obtained from two biological replicates. Statistical comparisons were performed using the unpaired two-tailed Student's $t$ test in (**b, d, e, h**) and the unpaired two-tailed Mann–Whitney $U$ test in (**f**) and (**g**). Source data are provided as a Source Data file.

(E-pep) of Synapsin to assess its potential functions. Injection of E-pep affected the kinetics of SV fusion and was suggested to play a role in the synchronicity of neurotransmitter release[18]. A recent study has shown that the E-domain of Synapsin 1 interacts with alpha-synuclein,

which can also undergo LLPS[60]. Interfering peptides of the E-domain used in that study and others have shown pronounced effects on the reserve pool of SVs[61]. Similarly, the role of two Synapsin domains was tested using antibodies targeting either the D- or E-domains[26]. It is

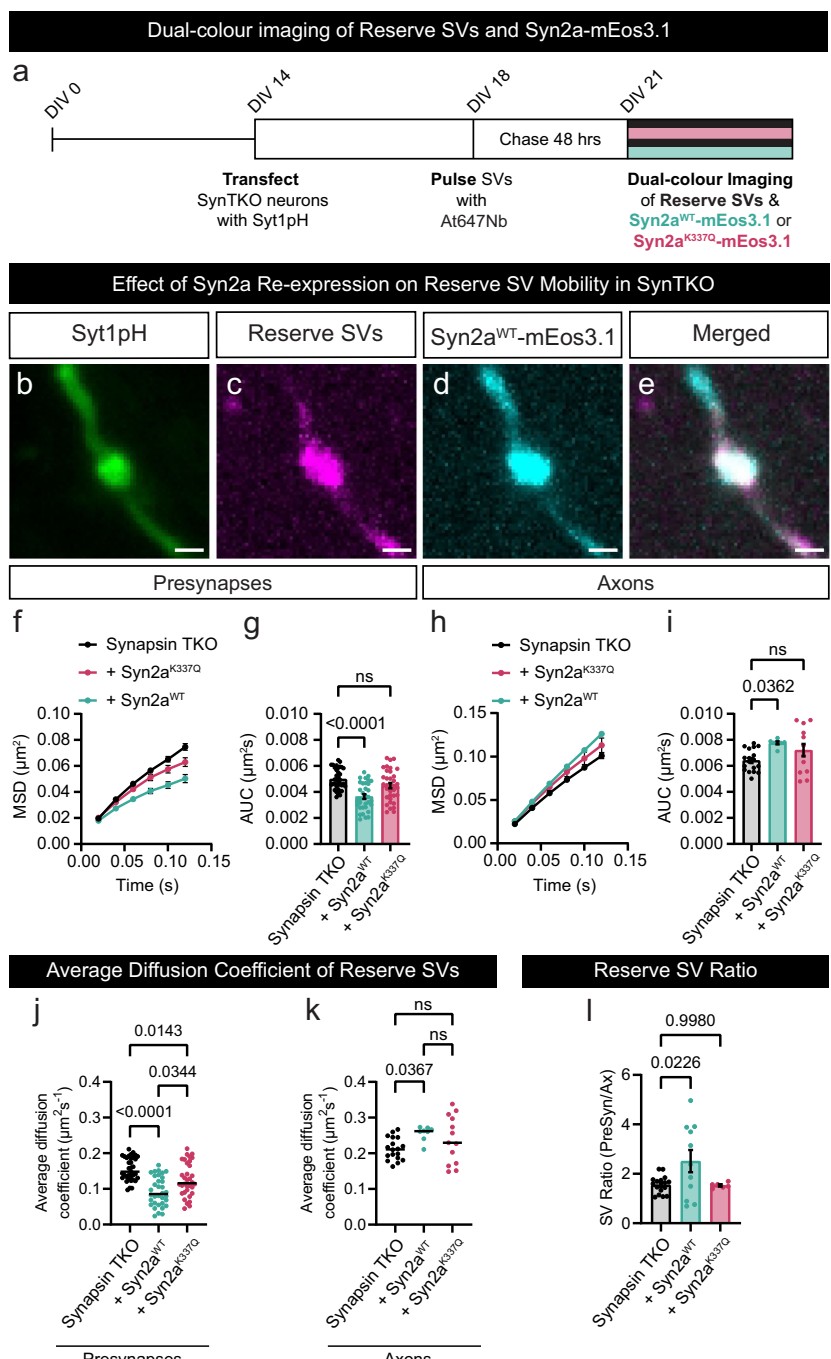

**Fig. 9 | Synapsin 2a^WT-mEos3.1 fully rescues the SynTKO reserve SV pool mobility phenotype. a** Graphical representation of the timeline for dual-colour single-particle tracking of the reserve pool of synaptic vesicles (SVs; black line) and either wild-type (WT) Synapsin 2a (Syn2a^WT-mEos3.1; cyan line) or the Synapsin 2a tetramerization deficient mutant (Syn2a^K337Q-mEos3.1; magenta line) in Synapsin triple knockout (SynTKO) hippocampal neurons. **b** Representative epifluorescence image of a neuronal segment expressing Synaptotagmin1-pHluorin (Syt1pH). **c**–**e** Maximum intensity projections of (**c**) Syt1pH-bound anti-green fluorescent protein (GFP) Atto647N nanobodies (Syt1pH/At647Nb; reserve pool) and (**d**) Syn2a^WT-mEos3.1. **e** Merged maximum intensity projections of (**c**) and (**d**). **f**, **h** Average mean square displacement (MSD; μm²) of reserve SVs in SynTKO neurons (Synapsin TKO; black), reserve SVs when Syn2a^K337Q-mEos3.1 is expressed (+Syn2a^K337Q magenta), and reserve SVs when Syn2a^WT-mEos3.1 is expressed (+Syn2a^WT; cyan), within the (**f**) presynapses and (**h**) axons of SynTKO hippocampal neurons. **g**, **i** Area under the MSD curve (AUC; μm²s) for the (**g**) presynapses and (**i**) axons of SynTKO hippocampal neurons. **j**, **k** Average diffusion coefficient of reserve SVs in SynTKO neurons (Synapsin TKO; black), reserve SVs when Syn2a^K337Q-

mEos3.1 is expressed (+Syn2a^K337Q; magenta), and reserve SVs when Syn2a^WT-mEos3.1 is expressed (+Syn2a^WT; cyan), within the (**j**) presynapses and (**k**) axons of SynTKO hippocampal neurons. **l** Ratio of reserve SVs detected in presynapses to reserve SVs detected in the axons in SynTKO neurons (black), SynTKO neurons when Syn2a^K337Q-mEos3.1 is expressed (magenta), and SynTKO neurons when Syn2a^WT-mEos3.1 is expressed (cyan). Data are displayed as mean ± SEM. Values were obtained from n = 36 presynapses (Synapsin TKO), n = 33 presynapses (+Syn2a^WT) and n = 35 presynapses (+Syn2a^K337Q) in (**f**, **g**), from n = 18 axons (Synapsin TKO), n = 7 axons (+Syn2a^WT) and n = 13 axons (+Syn2a^K337Q) in (**h**, **i**), from n = 35 presynapses (Synapsin TKO), n = 34 presynapses (+Syn2a^WT) and n = 34 presynapses (+Syn2a^K337Q) in (**j**), from n = 18 axons (Synapsin TKO), n = 8 axons (+Syn2a^WT) and n = 13 axons (+Syn2a^K337Q) in (**k**) and from n = 17 presynapses (Synapsin TKO), n = 11 presynapses (+Syn2a^WT) and n = 7 presynapses (+Syn2a^K337Q) in (**l**). Data were obtained from three biological replicates. Statistical comparisons were performed using the one-way ANOVA and Dunnett's or Tukey's multiple comparisons test in (**f**–**l**). Scale bar 1 μm (**b**–**e**). Source data are provided as a Source Data file.

important to note that the D-domain is not present in Syn2 and that the E-domain is only present in the 'a' isoforms of Synapsins (Synapsin 1a, 2a and 3a). The effect of the IgG antibody directed against the D-domain of Synapsin 1 was to disperse the cluster of SVs, by affecting IDR-IDR interactions. The antibody directed against the E-domain had a more subtle effect on the SV cluster, primarily by affecting the distal cluster following stimulation, suggesting that the E-domain was less critical in maintaining Synapsin BMCs. Synapsin isoforms have also been shown to form oligomeric structures both in vitro and in vivo[2,29,47]. Previous studies have identified key residues in the structural domain (C-domain) of Synapsin that mediate its assembly into tetramers, which are essential for SV-tethering. One missense mutation known to perturb tetramer assembly (K337Q), used here in our study, was unable to rescue both reserve SV mobility, and reserve SV density at the presynapse in SynTKO neuronal cultures. This suggests that homo-tetramerization of Syn2a is sufficient to rescue reserve pool mobility dynamics within the pre-synapse. In this study, the tetramerization-deficient Syn2a$^{K337Q}$ mutant displayed higher mobility compared to the Syn2a$^{WT}$ and its mobility was unaffected by stimulation in presynaptic hippocampal nerve terminals. This is in line with a recent study which demonstrated that Synapsin tetramerization promotes the formation of a densely interconnected network of SVs, an effect eliminated by the K337Q mutation[25]. This study also revealed that Syn2a tetramerization plays a critical role in clustering SVs in excitatory hippocampal synapses, but not in inhibitory synapses[25]. Although we could not differentiate excitatory from inhibitory neurons in our imaging system, the vast majority of neurons in these cultures are excitatory[62]. Thus, our results likely reflect the recycling and reserve pools of excitatory synapses. This points to the critical importance of tetramerization in the nanoscale organization of Synapsin and of the reserve pool of SVs in excitatory synapses. SVs residing in distal clusters were shown to be associated with electron-dense protein bridges, which appeared to link SVs to each other[31]. In the absence of Synapsins, these electron-dense bridges were less frequently observed than in WT conditions[31], and the SV cluster was dispersed[24]. Ultimately, hetero-tetramerization, and/or synergistic/opposing roles of Synapsin 1, in the background of Syn2a function, is likely to be important in regulating the SV cluster. It has been suggested that domain E (present in Synapsin 1a, 2a and 3a isoforms) may adopt a different conformation, potentially during oligomerisation, or within BMCs. In addition, such intramolecular interactions could regulate the targeting of Synapsin isoforms into different presynaptic or axonal sub-cellular compartments.

In summary, we have simultaneously imaged the reserve and recycling pools of SVs in live hippocampal neurons and have used this approach to obtain insights into their respective mobilities. Further, we have uncovered key nanoscale differences between the reserve and recycling pool and demonstrated, that the reserve pool of SVs dynamically extends beyond the presynaptic terminal into the adjoining axons. This contribution of the reserve pool to the superpool points to a potential role in synaptic plasticity previously restricted to recycling SVs[11]. Finally, we demonstrated that tetramerization of Syn2a controls the mobility of reserve pool SVs in live hippocampal cultures. The role of Synapsin in SV clustering, particularly its tetramerization, opens avenues for exploring synaptic function and plasticity. Questions remain about the mechanisms generating high-density regions and the long-range forces involved. Future studies should investigate how clustering processes for reserve and recycling SVs interact to regulate synaptic function/plasticity.

## Methods
### Ethical statement
All experimental procedures using animals were conducted under the guidelines of the Australian Code of Practice for the Care and Use of Animals for Scientific purposes and were approved by The University of Queensland (UQ) Animal Ethics Committee (2020/AE000439, 2020/AE000379 and 2023/AE000169) and Nanyang Technological University Institutional Animal Care and Use Committee (NTU-IACUC; described in the animal use protocol (AUP) A18095 / A21020). Mice used in this study were C57BL/6 strain wild-type (WT) and homozygous Synapsin triple knockout (SynTKO; B6;129-$Syn2^{tm1Pggd}$ $Syn3^{tm1Pggd}$ $Syn1^{tm1Pggd}$/Mmjax) mice[47]. WT mice were housed at the University of Queensland Biological Resources (UQBR). Mice were maintained in a 12-hour light/dark cycle and housed in a PC2 facility with ad libitum access to food and water. SynTKO mice were housed at Nanyang Technological University in accordance with approvals from NTU-IACUC. Both male and female mice were used for all experiments. SynTKO postnatal day 0 pups (P0) were euthanised, and hippocampi were dissected and shipped in hibernation buffer (BrainBits HEB500, #NC0427664) to The University of Queensland for subsequent culturing.

### Hippocampal neuronal culture
Primary hippocampal neurons were obtained from C57BL/6J embryonic day (E) 16, and SynTKO P0 mice. Isolated hippocampi were prepared as previously described[38]. Briefly, for live cell super-resolution microscopy, ~45,000 neurons were seeded onto poly-L-lysine (1 mg/mL Poly-L-lysine in Borate Buffer) coated 29 mm glass-bottom dishes with 10 mm micro-wells (Cellvis, #D29-10-1.5-N) in Neurobasal medium (Gibco, #21103-049) supplemented with 5% fetal bovine serum (Hyclone), 2% Glutamax (Gibco, #35050061) and 50 U/mL penicillin/streptomycin (Invitrogen, #15140122). The medium was changed to serum-free Neurobasal medium supplemented with 2% B27 (Gibco, #17504-044) 5–6 h post-seeding, and the medium was supplemented every week. Hippocampal neurons were grown until days in vitro (DIV) 22. Primary neurons were transfected at DIV14-15 using Lipofectamine® 2000 (Invitrogen, #11668019), as per manufacturer's instructions and imaged by super-resolution microscopy 3–7 days post-transfection.

### Plasmids
The synaptotagmin1-pHluorin (Syt1pH) plasmid was kindly provided by Prof. Volker Haucke (Freie Universität Berlin). The pmEos3.1-N1-Synapsin 2a WT (Synapsin 2a-mEos3.1) and pmEos3.1-N1-Synapsin 2a K337Q (Synapsin 2a-K337Q-mEos3.1) plasmids were kindly provided by Dr. Sang-Ho Song (Nanyang Technological University, Singapore), which were generated by subcloning either WT Synapsin 2a or the Synapsin 2a K337Q mutant into the pmEos3.1-N1 vector. The Synapsin 2a K337Q mutation was generated using the site-directed mutagenesis method using the following primers: (forward:5'-AGGGAACTGGCAGACAAACACTG-3', reverse:5'-GAGATGGATGTCCTCATG-3')[25]. The pmTagBFP-N1 plasmid was kindly provided by Prof. Vladislav Verkhusha[63].

### Super-resolution microscopy with oblique illumination
For live-cell super-resolution microscopy, transfected neurons were maintained at 37 °C with 5% $CO_2$ within the microscope stage. The super-resolution microscopy system consisted of an iLas2 ring-TIRF illumination system (Roper Scientific, France) mounted on a Nikon Ti-E inverted microscope with a Nikon CFI Apo TIRF 100x oil 1.49 NA objective and a Perfect Focus System (Nikon Corporation, Japan). Dual-colour imaging was performed using two Evolve 512 Delta EMCCD cameras (Teledyne Photometrics, USA) mounted on a TwinCam LS dual-emission image splitter (Cairn Research, UK), with a quad-band dichroic beam-splitter (ZT405/488/561/647rpc, Chroma Technology, USA), a triple-band emission filter on the reflected arm (ZET405/488/561 m, Chroma Technology, USA), and a far-red emission filter on the transmission arm (ET690/50 m, Chroma Technology, USA), allowing fine xy-alignment of both the transmitted and reflected ports. TetraSpeck™ microspheres (Invitrogen-Thermo Fisher Scientific, 0.1 µm) diluted in PBS were used to calibrate the correct alignment of the cameras.

## Dual pulse subdiffractional tracking of internalized molecules (DsdTIM), and dual colour super-resolution imaging

Cultured hippocampal neurons expressing Syt1pH were pulsed with a high K$^+$ Neurobasal medium (30 mM KCl, 2% Glutamax (Gibco), 50 U/mL penicillin/streptomycin (Invitrogen), 2% B27 (Gibco)) containing 400 pM anti-green fluorescent protein (GFP) Atto565-tagged nanobodies (At565Nb; GFP sdAb - FluoTag-Q, #N0301-At565-L, Synaptic Systems) for 1 min. After stimulation, unbound At565Nb were washed off with serum-free supplemented neurobasal medium, and neurons were chased for 48 h at 37 °C. After the chase, each dish was washed twice with a low K$^+$ imaging buffer (0.5 mM MgCl$_2$, 2.2 mM CaCl$_2$, 5.6 mM KCl, 145 mM NaCl, 5.6 mM D-glucose, 0.5 mM ascorbic acid, 0.1% BSA, and 15 mM HEPES, pH 7.4, 37 °C), and time-lapse movies (8,000 frames) were acquired at 50 Hz using MetaMorph software (version 7.7.8, Molecular Devices, CA, USA). After imaging the neurons in a resting state, neurons were then incubated with high K$^+$ imaging buffer (95 mM NaCl, 56 mM KCl, 2.2 mM CaCl$_2$, 0.5 mM MgCl$_2$, 5.6 mM D-glucose, 0.5 mM ascorbic acid, 0.1% BSA, 15 mM HEPES, pH 7.4, 37 °C) containing 200 pM anti-GFP Atto647N-tagged nanobodies (At647Nb; GFP sdAb - FluoTag-Q, #N0301-At647N-L, Synaptic Systems) for 5 min at 37 °C. After stimulation, unbound At647Nb were washed off with low K$^+$ imaging buffer, and neurons were then chased for 10 min at 37 °C. Electrical field stimulation was performed to validate the efficacy of the high K$^+$ stimulation protocol (Supplementary Fig. 4). To label the reserve pool of SVs, neurobasal medium containing 400 pM At565Nb was added to Syt1pH-transfected hippocampal neurons seeded in 35 mm glass-bottom dishes with 10 mm micro-wells (Thermo Scientific, #150680). The neurons were challenged with a train of 300 APs delivered at 50 Hz (100 mA and 1 ms pulse width) using a 35 mm dish insert with field stimulation electrodes (Warner Instruments, RC-21BRFS) and single channel digital stimulator (Panlab, #LE12106). After stimulation, unbound At565Nb were washed off with fresh neurobasal medium, and neurons were then chased for 48 h at 37 °C in their original conditioning medium.

## Single-particle tracking photoactivated localization microscopy (sptPALM)

Presynaptic compartments (axon and presynapses) from neurons positive for Syt1pH were selected using the green channel (~491 nm excitation). During imaging, neurons were incubated in a low K$^+$ imaging buffer and stimulated whilst in the microscope by incubating the neurons with the high K$^+$ imaging buffer. A 405 nm laser was used to photoconvert either Syn2a$^{WT}$-mEos3.1 or Syn2a$^{K337Q}$-mEos3.1, and a 561 nm laser was used simultaneously for excitation and bleaching of the resulting photoconverted single molecules. To spatially distinguish and temporally separate the stochastically activated molecules during the acquisition, the 405 nm laser was used between 1.5% and 5% of the initial laser power (100 mW Vortran Laser Technology), and the 561 nm laser was used at 70% of the initial laser power (150 mW Cobolt Jive).

## Single-particle trajectory analysis

The localization and tracking of single molecules were performed as previously described[38]. Briefly, single-molecule localizations were detected using wavelet-based segmentation, and trajectories were computed using a simulated annealing-based tracking algorithm with PALM-Tracer (version 2.1.0.28228), a custom-written software package tool that operates as a plugin of MetaMorph (Molecular Devices). Regions of interest (ROIs) were drawn around nerve terminals, defined as hotspots of increased pHluorin fluorescence[64,65]. Trajectories that lasted at least eight frames were reconstructed and the mean square displacement (MSD) was computed for each trajectory. The MSD was fitted by the equation MSD($t$)=$a$ + 4$Dt$ (where $D$=diffusion coefficient, $a$ = y intercept and $t$ = time). MSD (μm$^2$) is calculated and plotted over a 0.2 s time frame.

## Vesicular map reconstruction

We summarize here briefly the method we developed to analyze the single-particle trajectories SPTs and generate the various maps. We adopted a method[46] where the motion of vesicles is described by the Smoluchowski's limit of the Langevin equation. The position $X(t)$ satisfies the stochastic equation:

$$X = \frac{F(X(t),t)}{\gamma} + \sqrt{2D}\,\dot{W},\tag{1}$$

where $F(X)$ is a field of force that can restrict the motion of the vesicle due to the overall field, $W$ is a white noise, $\gamma$ is the friction coefficient[66] and $D$ is the diffusion coefficient. At a coarse spatiotemporal scale, we used the coarse-grained stochastic equation[43,67]:

$$\dot{X} = a(X) + \sqrt{2}B(X)\dot{W},\tag{2}$$

where $a(X)$ is the drift field and $B(X)$ the diffusion matrix. The effective diffusion tensor is given by $D(X) = \frac{1}{2}B(X)B^T(X)$ (Here $T$ denotes the transposition)[66]. The drift and diffusion fields from Eq. 2 can be recovered from single-particle trajectories acquired at any infinitesimal time step $\Delta t$ by estimating the conditional moments of the trajectory displacements $\Delta X = X(t + \Delta t) - X(t)$[66–70]:

$$a(x) = \lim_{\Delta t \to 0} \frac{E[\Delta X(t) \,|\, X(t)=x]}{\Delta t},\tag{3}$$

$$D(x) = \lim_{\Delta t \to 0} \frac{E[\Delta X(t)^T \Delta X(t) \,|\, X(t)=x]}{2\Delta t}.\tag{4}$$

The notation E[·|$X(t)$=$x$] represents averaging over all trajectories that are passing at point $x$ at time $t$. To estimate the local drift $a(X)$ and diffusion coefficients $D(X)$ at each point $X$ of the membrane and at a fixed time resolution $\Delta t$, we use a procedure based on a square grid.

The local estimators to recover the vector field and diffusion tensor[45] consist in grouping points of trajectories within a lattice of square bins $S(x_k, \Delta x)$ centered at $x_k$ and of width $\Delta x$. For an ensemble of $N$ two-dimensional trajectories $\{X_i(t_j) = (x_i^{(1)}(t_j), x_i^{(2)}(t_j)), i=1..N j=1..M_i\}$ with $M_i$ the number of points in trajectory $X_i$ and successive points recorded with an acquisition time $t_{j+1} - t_j = \Delta t$. The discretization of Eq. 3 for the drift $a(x_k) = (a^{(1)}(x_k), a^{(2)}(x_k))$ in a bin centered at position $x_k$ is

$$a^{(u)}(x_k) \approx \frac{1}{N_k} \sum_{i=1}^{N} \sum_{j=0, x_i(t_j) \in S(x_k, \Delta x)}^{M_i-1} \left( \frac{x_i^{(u)}(t_{j+1}) - x_i^{(u)}(t_j)}{\Delta t} \right),\tag{5}$$

where $u = 1.2$ and $N_k$ is the number of points $x_i(t_j)$ falling in the square $S(x_k, r)$. Similarly, the components of the effective diffusion tensor $D(x_k)$ are approximated by the empirical sums

$$D^{(u,v)}(x_k) \approx \frac{1}{N_k} \sum_{i=1}^{N} \sum_{j=0, X_i(t_j) \in S(x_k, \Delta x)}^{M_i-1} \frac{[x_i^{(u)}(t_{j+1}) - x_i^{(u)}(t_j)][x_i^{(v)}(t_{j+1}) - x_i^{(v)}(t_j)]}{2\Delta t}.\tag{6}$$

The centers of the bin and their size $\Delta x$ are free parameters that are optimized during the estimation procedure. To smooth the reconstructed diffusion and density maps and increase the accuracy, we weighted the points in the moving windows with a cosine function

as

$$a^{(u)}(x_k) \approx \frac{\sum_{i=1}^{N}\sum_{j=0,x_j(t_j)\in D(x_k,r)}^{M_i-1}\left(\frac{(x_i^{(u)}(t_{j+1})-x_i^{(u)}(t_j))}{\Delta t}w_{i,j}(x_k,r)\right)}{\sum_{i=1}^{N}\sum_{j=0,x_i(t_j)\in D(x_k,r)}^{N_s-1}w_{i,j}(x_k,r)} \qquad (7)$$

with $N_k$ the number of points of the trajectories falling in the disk $D(xk,r)$ of radius $r$ and centered at $x_k$. The weight of a displacement starting at $X_i(t_j)$ with respect to the disk $D(x_k,\Delta x)$ is given by

$$w_{i,j}(x_k,r) = cos\left(\frac{\pi}{2}\frac{||X_i(t_j)-x_k||}{r}\right), \qquad (8)$$

with $|.||$ the Euclidean norm. In that case, we use a refined grid $S(x_k,(\Delta x)')$ with bin size $(\Delta x)' = \Delta x/l_{sc}$, where $l_{sc}$ is a scaling factor. The role of the cosine weights $w$ is to decrease continuously the influence of the points falling near the boundary. Similarly, the generalized formula for the effective diffusion tensor $D(x_k)$ are given by the weighted sums

$$D^{(u,v)}(x_k) \approx \frac{\sum_{i=1}^{N}\sum_{j=0,x_i(t_j)\in D(x_k,r)}^{M_i-1}\frac{((x_i^{(u)}(t_{j+1})-x_i^{(u)}(t_j))(x_i^{(v)}(t_{j+1})-x_i^{(v)}(t_j)))w_{i,j}(x_k,r)}{2\Delta t}}{\sum_{i=1}^{N}\sum_{j=0,x_i(t_j)\in D(x_k,r)}^{M_i-1}w_{i,j}(x_k,r)},$$

where the weights $w$ are given by Eq. 8.

Finally, to estimate the local vesicle density, we computed the local density of points $\rho$ by using a procedure similar to the drift or diffusion estimation, dividing the image plane into a square bin $S(x_k,\Delta x)$. We then compute for each square of $S$ centered at $x_k$

$$\rho_{\Delta x}(x_k) = \frac{N_k}{(\Delta x)^2}, \qquad (9)$$

where $N_k$ is the number of trajectory points falling into the bin centered at $x_k$.

## Estimating Potential Well Parameters

High-density regions are described as a potential well due to a long-range field of force. The model is a basin of attraction obtained by truncated elliptic parabola with the associated energy function

$$U(X) = \begin{cases} A\left[\left(\frac{x^{(1)}-\mu^{(1)}}{a}\right)^2 + \left(\frac{x^{(2)}-\mu^{(2)}}{b}\right)^2 - 1\right], & X \in B \\ 0 & \text{otherwise} \end{cases} \qquad (10)$$

where $X = [x^{(1)},x^{(2)}]$, $\mu = [\mu^{(1)},\mu^{(2)}]$ is the center of the well, $a,b$ are the elliptic semi-axes lengths. The High-density region is approximated by the elliptic boundary defined as

$$B = \left\{X \text{ such that } A\left[\left(\frac{x^{(1)}-\mu^{(1)}}{a}\right)^2 + \left(\frac{x^{(2)}-\mu^{(2)}}{b}\right)^2 - 1\right] = 0\right\}. \qquad (11)$$

We estimated the various parameters such as the center, the boundary, the two axis, the parameter $A$ using estimators and hybrid algorithms as described previously[46]. The diffusion coefficient inside a well is considered to be constant and the energy of the well given by $E=A/D$ where $A$ is the attraction coefficient and $D$ the diffusion coefficient. The potential well model allows the estimation of the residence time using the escape time paradigm[43,66]: for a circular well, we have

$$\tau_e \approx \frac{Dr^2}{4A^2}e^{\frac{A}{D}}, \qquad (12)$$

with $r$ the radius of the well, $A$ its attraction coefficient and $D$ its diffusion coefficient. In the case of an elliptic well, we obtain an approximate circular boundary using the harmonic mean of the semi-axes $r = \sqrt{ab}$, where $a$ and $b$ are the large and the small-axes lengths respectively. This approximation holds for $a \approx b$.

## Statistical analysis

Results were analyzed statistically using GraphPad Prism software (GraphPad Software, Inc). The D'Agostino and Pearson test was used to test for normality. The unpaired two-tailed Student's $t$ test was used for the comparison of two groups, when the data were normally distributed. The non-parametric Mann–Whitney $U$ test was used when the data were not normally distributed. For datasets comparing more than two groups, we performed the Kruskal–Wallis or ordinary one-way ANOVA multiple comparison test. Statistical comparisons were performed on a per neuronal region (axon or presynapse) basis. All data points lying two standard deviations from the mean were considered outliers and removed from the dataset. These outliers were identified using the Outlier Wrangler custom-made Python script[42]. Primary hippocampal neurons were collected from at least two biological replicates. Values are represented as the mean ± SEM unless stated otherwise. The tests used are indicated in the respective figure legends. A $p$-value below 0.05 was considered significant and the numerical value was indicated in the graphs.

## Reporting summary

Further information on research design is available in the Nature Portfolio Reporting Summary linked to this article.

## Data availability

Single-particle trajectory data generated in this study are available for download from the publicly accessible institutional data repository of The University of Queensland (UQ eSpace) https://doi.org/10.48610/be3832e. Source data are provided with this paper.

## Code availability

Computer codes used to analyze the data have been described previously[42]. All Python codes used here are available at the GitHub repository https://github.com/tristanwallis/smlm_clustering. The NASTIC suite v1.0.7 is available at https://doi.org/10.5281/zenodo.10369046 under a Creative Commons CC BY 4.0 licence: you are free to use and modify the code on the proviso that you make any changes freely available, acknowledge the original authors in derivative works and do not release said works under a more restrictive licence.

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

## Acknowledgements

The super-resolution imaging was carried out at the Queensland Brain Institute's (QBI's) Advanced Microscopy Facility, led by Dr. Rumelo Amor and his team. We thank the QBI Information technology (IT) facility members and the entire QBI University of Queensland's Biological Resources (UQBR) animal team for their ongoing technical assistance with our projects. We thank Dr. Alex McCann for critically appraising and editing the manuscript. We acknowledge Prof. Volker Haucke for providing the Synaptotagmin1-pHluorin plasmid, Prof. Vladislav Verkhusha for providing the pmTagBFP-N1 plasmid, Dr Sang-Ho Song for providing the Synapsin 2a$^{WT}$-mEos3.1 and Synapsin 2a$^{K337Q}$-mEos3.1 plasmids, Karen Chung for providing synapsin TKO mice and Melissa Yeow (Nanyang Technical University, Singapore) for dissecting the SynTKO hippocampi. This work was supported by an Australian Research Council Discovery Project grant (ARC) (DP230102278), a National Institute On Aging of the National Institutes of Health (NIH) grant R21AG080435, an ARC Linkage Infrastructure, Equipment, and Facilities grant (LE130100078), and a National Health and Medical Research Council (NHMRC) Fellowship (1155794) awarded to F.A.M., as well as grant OFIRG/MOH-000225-00 from the Singapore National Medical Research Council to G.J.A. R.M.M. was supported by a Clem Jones Foundation Fellowship, the University of Queensland (UQ) Research Stimulus Allocation 2 fellowship and the NHMRC Boosting Dementia Research Initiative. S.F.L. was supported by a Research Training Program (RTP) Scholarship. M.J. was supported by a UQ Amplify Fellowship, UQ Early Career Research Grant (2057309) and the Australian Research Council Discovery Early Career Researcher Award (DE190100565). D.H. was funded by the European Research Council (ERC) under the European Union's Horizon 2020 research and innovation program (grant agreement No 882673), and ANR AstroXcite.

## Author contributions

F.A.M., G.J.A., M.J. and R.M.M. were responsible for the conceptualization and designed aspects of the study; S.F.L. was involved in setting up all methodological procedures required during this investigation; S.F.L. performed all super-resolution imaging; S.F.L. analyzed the super-resolution data; M.F., P.P., J.R. and D.H. performed the high-density region trajectory analysis; F.A.M. was involved in funding acquisition; F.A.M. led the project administration; R.M.M., G.J.A. and F.A.M. supervised the research; R.S.G., T.P.W., S.F.L. and F.A.M. wrote the original draft of the paper; S.F.L., R.S.G., R.M.M. and F.A.M. wrote, reviewed and edited the final version of the paper with input and substantial revisions from all authors.

## Competing interests

The authors declare no competing interests.
