## [Peer Review File · Nature Communications]

Synapsin 2a tetramerisation selectively controls the presynaptic nanoscale organisation of reserve synaptic vesiclesREVIEWER COMMENTS

Reviewer #1 (Remarks to the Author):

Using dual-pulse sub-diffractive Tracking of Internalised Molecules (DsdTIM) to simultaneously track SVs from the recycling and reserve pools, Longfield et al studied the mobility of SVs tagged by two fluorophores in the axon and boutons of cultured hippocampal neurons. They found that SVs in the reserve pool shows a lower mobility than those in the recycling pool in boutons, but similar mobility in axons. The mobility of SVs in the reserve pool is preferentially elevated in triple knockout of synapsins (Synapsin 1-3 TKO). This phenotype can be rescued by reintroduction of WT Synapsin2a but its K337Q mutant devoid of the tetramerization failed to do so. These observations led the authors to conclude that Synapsin2a tetramerization allows presynaptic association of SV clustering to reduce their mobility in the reserve pool of presynaptic boutons.

Overall this study provide some interesting observations on the dynamics of SVs in two pools distributed in boutons and axons respectively. The results are acquired with advanced optic technology and innovative imaging and analysis paradigms, yielding solid results. However, the conceptual novelty of this study is rather marginal, given that synapsin2a tetramerization has already been highlighted but the other referred paper. The author may want to consider the perturbations that promote or inhibit tetramerization to give this paper new twist.

Major comments:

1. It is not entirely clear what is the fraction of synaptic vesicles being labeled with K⁺ loading and tracked in real time? Although it is a valid approach, it does raise some concern if such approach may yield results that are not necessarily relevant to more physiological stimuli such as field stimulation. Prolonged K⁺ depolarization may label different sub-pools of SVs and even induce activity-dependent plasticity that does not reflect the mobile dynamics of SVs among different pools under physiological conditions.
2. In Fig 6-7, Re-introduction of syn 2a WT seem to have over-suppressed the motility of reserved pool while its mobility with K337Q mutant was more comparable to that of WT (MSD values in Fig 3-4), raising the question how meaningful such experiments are or whether these are artifacts from over-expression.
3. Synapsin 1-3 may form oligomers in native synapses so it might be too hasty to conclude that tetramerization of synapsin 2a is the core mechanism for presynaptic nanocluster of SVs. In fact, it would be ideal to use an approach to acutely perturb tetramerization, e.g. peptide, to destabilise tetramer or other forms of oligomerization in WT synapses to strengthen such a conclusion.

Reviewer #2 (Remarks to the Author):

Longfield et al present an advanced optical microscopy study on the live-cell dynamics of synaptic vesicles (SV) and their modulation by the protein synapsin2a. Through an optimized single-particle tracking algorithm, the authors were able to differentiate for the first time between recycling and reserve pools of SVs and how these are influenced by synapsin2a and especially the latter's the multimerization. This is a very clever and impressive approach with interesting results, which in principle deserve publication. Unfortunately, I find the story at parts difficult to follow due to several reasons, mainly since detailed explanations are missing.

- I was not convinced that the authors really proved that they could distinguish between the two different pools by just changing the temporal pattern of the labelling protocol. How can they be completely sure that the SVs are really 100% recycling or reserve? Maybe I did not get the details here.
- Concrete quantifications of mobility are missing throughout the text – maybe state comparing numbers of diffusion coefficients?
- The authors should discuss their results in light of the LLPS model. Can any links be drawn

between both models, e.g. between multimerization and condensation? Can it not be interpreted in both ways or congruently?

- What does AUC give us other than a qualitative measure of general mobility? Any quantitative interpretation?

- The authors included some controls to check for biasing diffusion of unbound/free At565Nb.

Maybe include more controls like the dye alone without Syt1pH?

- End page 8, beginning page 9: I got confused here. On one hand the authors state that the absence of synapsin did not dramatically alter the mobility patterns, on the other hand they state that synapsins selectively lower the mobility of the reserve pool. A slight rephrasing and indication of concrete numbers (diffusion coefficients) would help.

- Figure 5: This quantification must include the labelling degree, otherwise it might be biased. Any comment/control?

- Page 10, middle: What stimulation? Name it.

- Page 10, middle: What is meant by low MSD?

- Page 10, middle: What is meant by "regulates the nanoscale organisation"? Did not get the argument here.

Point-by-point response to referees

Reviewer #1 (Remarks to the Author):

Using dual-pulse sub-diffractive Tracking of Internalised Molecules (DsdTIM) to simultaneously track SVs from the recycling and reserve pools, Longfield et al studied the mobility of SVs tagged by two fluorophores in the axon and boutons of cultured hippocampal neurons. They found that SVs in the reserve pool shows a lower mobility than those in the recycling pool in boutons, but similar mobility in axons. The mobility of SVs in the reserve pool is preferentially elevated in triple knockout of synapsins (Synapsin 1-3 TKO). This phenotype can be rescued by reintroduction of WT Synapsin2a but its K337Q mutant devoid of the tetramerization failed to do so. These observations led the authors to conclude that Synapsin2a tetramerization allows presynaptic association of SV clustering to reduce their mobility in the reserve pool of presynaptic boutons.

Overall this study provide some interesting observations on the dynamics of SVs in two pools distributed in boutons and axons respectively. The results are acquired with advanced optic technology and innovative imaging and analysis paradigms, yielding solid results.

However, the conceptual novelty of this study is rather marginal, given that synapsin2a tetramerization has already been highlighted but the other referred paper. The author may want to consider the perturbations that promote or inhibit tetramerization to give this paper new twist.

Major comments:

1. It is not entirely clear what is the fraction of synaptic vesicles being labelled with K⁺ loading and tracked in real time? Although it is a valid approach, it does raise some concern if such approach may yield results that are not necessarily relevant to more physiological stimuli such as field stimulation. Prolonged K⁺ depolarization may label different sub-pools of SVs and even induce activity-dependent plasticity that does not reflect the mobile dynamics of SVs among different pools under physiological conditions.

The reviewer is correct that high potassium stimulation does not mimic all the events that occur during physiological stimulation, as it elicits high-frequency spontaneous release likely to promote various types of endocytosis. However, it is considered a well-established paradigm to elicit neurotransmitter release, used for many years in a multitude of neuronal and neurosecretory cell types. As requested, we have now successfully labelled the reserve pool of SVs using a more physiological stimulation paradigm – high-frequency field stimulation (50 Hz, 300 AP). To investigate the effect of electrical stimulation on the reserve pool of SVs and confirm the efficacy of our labelling technique, our lab had

to introduce an entirely new stimulation procedure. This involved importing specialised equipment, building the system, adapting it to our super-resolution acquisition procedure, and solving a multitude of technical and methodological problems. In addition, to successfully label and image the reserve pool of SVs, we had to adapt our new set-up to the sterile conditions needed to avoid contamination and preserve neuronal culture viability across the experimental duration.

These new experiments are now included in the new Supplementary Figure 4. These experiments demonstrate that there is no difference in the mean square displacement (MSD) between the high potassium and electrical stimulation paradigms 48 hours after labelling. This analysis was carried out both in the presynapses and axons and no differences were found in either compartment.

Supp Fig.4

Supplementary Fig 4. Comparison of reserve SV mobility 48 hours after being labelled with either high K⁺ or electrical field stimulation. (a, c) Average MSD of reserve SVs labelled using either a high K⁺ buffer (blue) or train of 300 APs (50Hz for 6s; orange), within the (a) presynapses and (c) axons. (b, d) Area under the MSD curve (AUC; $\mu\text{m}^2 \text{s}$) for (b) presynapses and (d) axons. Data are displayed as mean \pm SEM. Values were obtained from $n \geq 15$ (presynapses) and $n \geq 7$ (axons) from at least 7 neurons in a to d. Data was obtained from 1 independent neuronal culture. Statistical comparisons were performed using Student's *t* test or Mann–Whitney *U* test.

We have also amended both the Method and Results sections to highlight these results:

Methods:

'Electrical field stimulation was performed to validate the efficacy of the high K⁺ stimulation protocol (Supplementary Fig 4). To label the reserve pool of SVs, neurobasal medium containing 400 pM At565Nb was added to Syt1-pH transfected hippocampal neurons seeded in Nunc™ 35 mm glass-bottom dishes with 10 mm micro-wells (Thermo Scientific, #150680). The neurons were challenged with a train of 300 action potentials (APs) delivered at 50Hz (100 mA and 1 ms pulse width) using a 35mm dish insert with field stimulation electrodes (RC-21BRFS; Warner Instruments, Holliston, MA) and single channel digital stimulator (Warner Instruments, Holliston, MA, Panlab, #LE12106). After stimulation, unbound Atto565Nbs were washed off with fresh neurobasal medium, and neurons were then chased for 48 hrs at 37°C in their original conditioning medium.'

Results:

'Lastly, to test whether the depolarising stimulus (high K⁺) captured the dynamics of the reserve pool vesicles, we tested a more physiological stimulation paradigm. We performed high-frequency field stimulation (50 Hz, 300 action potentials (AP)) in sterile conditions in mature hippocampal neurons expressing Syt1pH (as above) in the presence of anti-GFP-At647Nbs and chased for 48h. The mobility of the labelled vesicles was indistinguishable from those observed following high K⁺ stimulation (Supplementary Fig. 4), thereby validating our high K⁺ stimulation protocol.'

2. In Fig 6-7, Re-introduction of syn 2a WT seem to have over-suppressed the motility of reserved pool while its mobility with K337Q mutant was more comparable to that of WT (MSD values in Fig 3-4), raising the question how meaningful such experiments are or whether these are artifacts from over-expression.

Figures 6 and 7 do not relate to the mobility of the reserve pool of synaptic vesicles. They examine the mobility of Synapsin 2a-mEos2 itself in resting and stimulated conditions. The reviewer may have been referring to Figure 8, where the mobility of the reserve pool of vesicles was investigated in SynTKO neurons following rescue conditions. During the revision of this manuscript, we realised that these data were underpowered. As a result, we imported more SynTKO hippocampi and repeated these experiments (increasing the number of neurons analysed). We have now also included a new set of analyses focusing on the mobility of reserve SVs within the axons (Figure 9h, i). Importantly, we found that the mobility of axonal reserve SVs in neurons rescued with KQ mutant was equivalent to SynTKO. See the figure excerpt below:

Fig 9. Syn2a^{WT}-mEos3.2 fully rescues the reserve SV pool mobility SynTKO phenotype. (f, h) Average MSD of reserve SVs in SynTKO neurons (black), reserve SVs when Syn2a^{K337Q}-mEos3.2 is expressed (magenta), and reserve SVs when Syn2a^{WT}-mEos3.2 is expressed (cyan), within the (f) presynapses and (h) axons of SynTKO hippocampal neurons. (g, i) Area under the MSD curve (AUC; $\mu\text{m}^2\text{s}$) for the (g) presynapses and (i) axons of SynTKO hippocampal neurons. (j, k) Average diffusion coefficient of reserve SVs in SynTKO neurons (black), reserve SVs when Syn2a^{K337Q}-mEos3.2 is expressed (magenta), and reserve SVs when Syn2a^{WT}-mEos3.2 is expressed (cyan), within the (j) presynapses and (k) axons of SynTKO hippocampal neurons. (i) Comparisons of the ratio of detections in the axons and presynapses from reserve SVs in SynTKO neurons (black), reserve SVs when Syn2a^{K337Q}-mEos3.2 is expressed (magenta), and reserve SVs when Syn2a^{WT}-mEos3.2 is expressed (cyan). Data are displayed as mean \pm SEM. Values were obtained from $n \geq 33$ (presynapses) and $n \geq 7$ (axons) from at least 12 neurons in f

and i. Data was obtained from ≥ 3 biological replicates. Statistical comparisons were performed using one-way ANOVA and Dunnett's or Tukey's multiple comparisons test.

The reviewer is right that the effect of Synapsin 2a^{WT} re-expression may be stronger than expected. Our interpretation of this response is that as Syn2a is the only isoform able to fully rescue the SynTKO phenotype in excitatory neurons (Gitler *et al.*, 2008), it likely overcompensates when expressed in SynTKO in isolation. It is important to note that this effect was much reduced with the tetramerisation mutant, suggesting a key role of synapsin tetramerisation in immobilising the reserve pool of vesicles. We have now included a discussion paragraph on the potential consequences of synapsin homo- and hetero-dimers.

Discussion:

'Synapsins have been shown to form oligomeric structures both in vitro and in vivo^{22,48,68}. Synapsins structural domain (C-domain) mediate its assembly into tetramers. Previous studies have identified key residues in the C-domain of synapsin that mediate its assembly into tetramers, that are essential for SV-tethering. One missense mutation known to perturb tetramer assembly, used in our study (K337Q) was unable to rescue reserve vesicle mobility and the SV density at the presynapse in SynTKO neuronal cultures. This suggests that homo-tetramerization of Syn2a is sufficient to rescue reserve pool mobility dynamics within the pre-synapse. Hetero-tetramerization, and/or synergistic/opposing roles of Synapsin1, in the background of Syn2a function, is likely to be important in regulating the SV cluster. It has been suggested that domain E (present in Synapsin1a, 2a and 3a isoforms) may adopt a different conformation, potentially during oligomerisation or within BMCs. In addition, such intramolecular interactions could regulate the targeting of Synapsin isoforms into different presynaptic or axonal sub-cellular compartments.'

3. Synapsin 1-3 may form oligomers in native synapses so it might be too hasty to conclude that tetramerization of synapsin 2a is the core mechanism for presynaptic nanocluster of SVs. In fact, it would be ideal to use an approach to acutely perturb tetramerization, e.g. peptide, to destabilise tetramer or other forms of oligomerization in WT synapses to strengthen such a conclusion.

It is unquestionable that synapsin can generate biomolecular condensates (BMCs) *in vitro* and in neurons (Zhang and Augustine, 2021; Gitler *et al.*, 2004; Hosaka and Südhof, 1999). However, the mechanism by which the reserve pool of synaptic vesicles is immobilised in nerve terminals is still an open question. Our results suggest that the tetramerization of Synapsin 2a could be involved in the cross-linking of synaptic vesicles. It is still unclear how this mechanism only targets the reserve pool of

SVs and whether synapsin tetramerization is involved in modulating the ability of other Synapsins to undergo phase separation, forming BMCs in the synapse. More work is therefore needed to distinguish between these two possibilities.

We thank the reviewer for their suggestions. The mutant form of Synapsin 2a (Syn2a^{K337Q}) that we used has been reported to selectively prevent the tetramerization of this Synapsin (Gitler et al, 2008; Song and Augustine, 2023). We do not currently have tools to acutely perturb oligomerization. We agree with the reviewer that our results should be interpreted carefully, in light of various recent papers including (Hoffmann et al., 2023; Song and Augustine, 2023). We have therefore, carefully amended the discussion to reflect this discrepancy in our interpretation and to lay the basis for future studies.

Discussion:

“Synapsin isoforms have been shown to form oligomeric structures both in vitro and in vivo^{22,48,68}. Synapsins structural domain (C-domain) mediate its assembly into tetramers. Previous studies have identified key residues in the C-domain of synapsin that mediate its assembly into tetramers, that are essential for SV-tethering. One missense mutation known to perturb tetramer assembly, used in our study (K337Q) was unable to rescue reserve vesicle mobility and the SV density at the presynapse in SynTKO neuronal cultures. This suggests that homo-tetramerization of Syn2a is sufficient to rescue reserve pool mobility dynamics within the pre-synapse. Hetero-tetramerization, and/or synergistic/opposing roles of Synapsin1, in the background of Syn2a function, is likely to be important in regulating the SV cluster. It has been suggested that domain E (present in Synapsin1a, 2a and 3a isoforms) may adopt a different conformation, potentially during oligomerisation or within BMCs. In addition, such intramolecular interactions could regulate the targeting of synapsin isoforms into different presynaptic or axonal sub-cellular compartments.”

Reviewer #2 (Remarks to the Author):

Longfield et al present an advanced optical microscopy study on the live-cell dynamics of synaptic vesicles (SV) and their modulation by the protein synapsin2a. Through an optimized single-particle tracking algorithm, the authors were able to differentiate for the first time between recycling and reserve pools of SVs and how these are influenced by synapsin2a and especially the latter's multimerization. This is a very clever and impressive approach with interesting results, which in principle deserve publication. Unfortunately, I find the story at parts difficult to follow due to several reasons, mainly since detailed explanations are missing.

We thank the reviewer for their positive remarks about our approach. We have now increased the critical details of our methodology to improve the readability and interpretation of our manuscript.

- I was not convinced that the authors really proved that they could distinguish between the two different pools by just changing the temporal pattern of the labelling protocol. How can they be completely sure that the SVs are really 100% recycled or reserve? Maybe I did not get the details here.

Our protocol is based on previously published work done by Truckenbrodt *et al.* (2018). Here, the authors studied the aging of synaptic vesicle proteins using antibodies directed against the luminal domain of synaptotagmin 1 (syt1) to reveal the localization and activity status of vesicle molecules over time. They showed that (1) antibody-labelled syt1 proteins gradually decrease in synapses over several days (0-10 days with 48hrs standing out as a key time point). (2) Antibody-labelled syt1 molecules become rapidly inactive in terms of exocytosis over time, that many labelled molecules still remain within synapses but do not participate in release under normal network activity. Further, that these molecules can be induced to release by strong stimulation. Finally, (3) Newly synthesized proteins are preferentially incorporated into actively recycling vesicles compared to inactive vesicles and that inactivated vesicles do not return to the actively recycling pool under normal network activity. Data all suggesting that as syt1 SV proteins age, they incorporate into a population of vesicles that are no longer fusogenic under normal network activity and do not co-localise with new recycling vesicles (aka the reserve pool).

As all the work done by Truckenbrodt *et al.* (2018) was carried out in fixed preparations, we were able to expand on this research in our investigation of SV incorporation into the reserve pool of live hippocampal neurons (Figure 1). Our results clearly show a significant and progressive decrease in the mobility of labelled vesicles following longer chasing times, as expected from the reserve pool. Further,

we assessed the mobility of this vesicular pool in response to restimulation and, as expected from the reserve pool, no change in mobility was detected. This suggests that they do not respond to stimulation and undergo fusion with the plasma membrane as previously found with the recycling pool of SVs (Joensuu et al., 2016; Figure 3a-f).

Fig.1

Fig. 1. Tracking the reserve pool of SVs in live hippocampal neurons. (a) Graphical representation of the sdTIM protocol optimization for the reserve pool of SVs: DIV19 hippocampal neurons expressing Synaptotagmin1-pHluorin (Syt1pH) were stimulated with high K^+ medium containing anti-GFP Atto565-tagged nanobodies (At565Nbs) (red) for one minute. Following stimulation, the excess

nanobodies were washed off, and the neurons were chased for various time periods in conditioning medium as indicated in the figure, and then imaged in a low K^+ imaging buffer. **(b-e)** Representative epifluorescence Syt1pH images (b, d) and corresponding Syt1pH/At565Nb SV trajectory maps (c, e) (colour coded by their instantaneous diffusion coefficients; colour bar represents $\text{Log}_{10}[\mu\text{m}^2\text{s}^{-1}]$) of presynapses chased for **(b, c)** 10 min and **(d, e)** 48 hrs. **(f, h)** Average MSD of the trajectories generated from Syt1pH/At565Nb tracks in the **(f)** presynapse and **(h)** axons at different time points. **(g, i)** Area under the MSD curve (AUC; $\mu\text{m}^2\text{s}$) at **(g)** presynapses and **(i)** axons. Data are displayed as mean \pm SEM. Values were obtained from the presynapses and axonal segments of at least $n \geq 13$ neurons per condition in f to i. Data was obtained from ≥ 3 independent neuronal cultures. Statistical comparisons were performed using one way ANOVA and Dunnett's or Tukey's multiple comparisons test. The different chase time points in f to i were compared to the 10 min chase time point, which was considered as recycling SV mobility control. Scale bars 1 μm (b-e).

Fig.3

Fig 3. Quantification of the reserve and recycling pool mobilities. (a, c) Average MSDs of the resting reserve pool (RP) of SVs (black), RP SVs after stimulation (red) and the recycling pool of SVs (green) within the (a) presynapses and (c) axons. (b, d) Area under the MSD curve (AUC; $\mu\text{m}^2 \text{ s}$) for (b) presynapses and (d) axons. (e, f) Average diffusion coefficient of the RP (black), RP SVs after stimulation (red) and the recycling pool of SVs (green) within the (e) presynapses and (f) axons. (g, h) Comparisons of the density of detections the axons and presynapses from (g) recycling and (h) reserve vesicles tagged with anti-GFP At565Nbs or At647Nbs respectively, normalised by the area ($\text{traj}/\mu\text{m}^2$). Data are displayed as mean \pm SEM. Values were obtained from $n \geq 28$ (presynapses) and $n \geq 11$ (axons) from at

least 15 neurons in panels a to h. Data was obtained from ≥ 3 independent neuronal cultures. Statistical comparisons were performed using one way ANOVA and Dunnett's or Tukey's multiple comparisons test.

Additional context as to the choice of pulse chase intervals has been included in the Results section and we have more clearly referred the important body of work that has helped shape our new experimental paradigm.

Results:

'These images qualitatively show the lower mobility of Syt1pH/At565Nb trajectories at 48 hrs, solidifying the importance of this time point in SV maturation, and complimenting previous Syt1 experiments performed by Truckenbroldt et al, (2018). In this study the authors demonstrated that Syt1 labelled 48 hours earlier, no longer co-localised with newly labelled SV proteins and were no longer fusogenic, suggesting that these 'old' Syt1-tagged vesicles were segregated from the recycling pool and transitioned into the reserve pool⁴³. Further quantification of all time points showed a progressive decrease in SV mobility over time, specifically within the presynaptic terminal (Fig.1f, g), but not in the axonal compartment (Fig.1h, i). The lowest SV mobility in the presynaptic compartment was exhibited at 48 hrs, confirming that by this time recycling SVs had transitioned into the reserve pool.'

To further investigate the differences between the reserve and recycling population of SVs we have set up a collaboration with David Holcman's Group (Group of Data Modelling and Computational Biology, IBENS, Ecole Normale Supérieure, 75005 Paris, France) to perform high-throughput statistical analysis on our data. This approach is based on computing diffusion and density maps as well as identifying a high-density regions of SV trajectories in live hippocampal neurons. Here, we found that the reserve pool of SVs had a significantly altered probability of generating sub-micron regions that act as 'sinks' with higher energy (kT) and residence time. These new data are presented in the new Figure 4 and we have included our interpretation of this information in our Results and Discussion sections.

Fig.4

Fig. 4 Density and diffusion maps describing the differences between reserve recycling SVs. (a, b) Representative neuron where recycling (a) and reserve (b) SVs displayed as individual trajectories. (c, d) Density map of recycling (c) and reserve (d) SVs. (e, f) Diffusion maps of recycling (e) and reserve (f) SVs where the high-density regions are delimited by an ellipse (red). (g-j) High density regions of recycling (g, h) and reserve (i, j) SVs are characterized by converging arrows (4 colours corresponding to 4 main directions). The ellipse corresponds to the boundary found by automated algorithms⁴⁷ and the energy of the wells is in the unit of kT. (k) Energy (in kT) of the associated recycling and reserve SV potential wells. (l) Residence time (s) inside a potential well of the associated recycling and reserve SVs. Data are displayed as mean \pm SEM. Values were obtained from $n = 4$ (recycling pool potential wells) and $n = 77$ (reserve pool potential wells) from at least 5 neurons in panels k and l. Data was obtained from ≥ 3 independent neuronal cultures. Statistical comparisons were performed using Kolmogorov Smirnov test.

Results:

To further characterize the differences between recycling and reserve pools of vesicles, we used a high throughput statistical approach based on computing the diffusion and density maps as well as identifying the high-density regions⁴⁴⁻⁴⁶. The underlying biophysical model (equation 2 in Methods) assumes that vesicles can either move according to Brownian motion or interact with the local environment that can stabilize them in given sub-micrometre subregions. The density and diffusion maps are constructed from the ensemble of trajectories by estimating the local density and diffusion coefficients in a grid map decomposed into small bins (see Methods for the statistical estimators eqs. 9-5). The results are shown in Fig. 4a-j. The density map reveals areas (red) characterized by local high-density regions (HDRs) the recycling and reserve SV populations of live hippocampal neurons (Fig. 4g-j illustrates examples of potential well or SV traps in these pools). The diffusion map reveals a more uniform distribution of SV mobility within the axonal compartment (Fig. 4e-f). To analyse these HDRs, we use the potential paradigm framework⁴⁴ under which such regions result from long-range force interaction. We first observed that high-density regions are much more frequent for reserve ($n=77$) than recycling vesicles ($n=4$) for the same neuronal region.

We further characterized these HDRs by extracting their boundaries and associated energy using an automated classification algorithm⁴⁷. We report here that the size of the wells associated with the reserve pool was larger ($0.18 \pm 0.07 \mu\text{m}$) than those for recycling ($0.1 \pm 0.01 \mu\text{m}$) (Supplementary Fig. 5). In addition, the stability of the reserve pool measured by the energy of the well shows that it is higher ($E = 2.15\text{kT}$) than for recycling ($E = 0.94\text{kT}$) (Fig 4k). Lastly, we determined the residence time in confinement versus Brownian diffusion of trajectories within the wells. We found that the potential

wells were able to confine reserve SVs for at least 3x times longer than SVs of the recycling pool (Fig. 4l-m). The reserve pool of SVs is therefore characterized by a much larger number of high-density regions defined as potential wells. These wells are larger in size (Supplementary Fig. 5) and their energy is also larger compared to that of the recycling pool (Fig. 4k). This (1) confirms that the two populations of DsdTIM labeled SVs are indeed different (as they display unique mobility patterns) and (2) suggests that the reserve pool is much more stable, and the associated mechanism of SV trapping involves a greater force than that of the recycling pool.'

Discussion:

'In depth analysis of single SV trajectories reveals that the most striking difference between the reserve and recycling pools stemmed from the number and size of high-density regions, that were much more prominent for the reserve pool SVs. These high density regions were both larger and more stable for the reserve pool, suggesting that the two types of vesicles interact differently with their nanoscale presynaptic environment. Surprisingly, high density regions were also found along the axon which could indicate either that these are silent synapses⁵⁸ and/or that the reserve pool has an intrinsic ability to generate clusters in axons. The analysis of the reserve pool SV trajectories in synapsin TKO neurons revealed a clear role of synapsin in shaping both the size of these high density regions and the strength of the interactions between vesicles of the reserve pool. It remains unclear how these high-density regions are generated and what mechanism(s) determine their size, which can extend for up to two hundred nanometres. In particular, it is unclear how these long-range forces are generated via synaptic protein self-assembly mechanisms and/or interactions with cytoskeletal elements of the presynapse. How the distinct mechanisms underpinning the clustering of reserve and recycling SV pools cross-talk to filter recycling SV to the active zone and restrict the reserve vesicles from accessing this critical zone will require further investigation.

Further, we established the role of synapsin 2a tetramerization in dynamically anchoring the reserve pool of SVs at the presynapse. The conventional understanding of how SVs cluster at the presynapse is currently under scrutiny, particularly as we make the technological advances necessary to study protein interactions at this scale. Ultrastructural analysis of SynTKO neuronal synapses revealed dispersed SVs and altered SV clusters adjacent to the active zone²⁴. Synapsins were shown to undergo LLPS and mediate membraneless compartments called biomolecular condensates (BMCs) at the presynapses²⁴. Synapsin BMCs could underpin the formation of the potential wells described herein and characterized by long-range interactions.'

- Concrete quantifications of mobility are missing throughout the text – maybe state comparing numbers of diffusion coefficients?

We thank the reviewer for their suggestion. We have now analysed the diffusion coefficients throughout the manuscript in Figures: Fig. 3e, f; Fig. 4 and Fig. 9j, k.

And included descriptions of the data in the Results section:

'We quantified the mobility of the reserve and recycling SV pools within the presynaptic and axonal compartments using the MSD (Fig. 3a-d) and average diffusion coefficients (Fig. 3e-f) of the Syt1pH/At565Nb and At647Nb trajectories.'

'We further assessed this data, extracting the average diffusion coefficient of the reserve and recycling SVs in both the presynapses and axons (Fig. 3e, f), which showed the same mobility trends as the MSD and AUC data, with the reserve and recycling pools displaying significantly different mobilities and the mobility of the reserve pool remaining unaffected by stimulation.'

See Fig 4 results section above.

'Further, we assessed the diffusion coefficient of reserve SVs in both the presynapses and axons (Fig. 9j, k). We found that re-expression of Syn2a^{WT}-mEos3.2 in SynTKO dramatically reduces the mobility, an effect which was significantly less pronounced for the Syn2a^{K337Q}-mEos3.2 at the presynapse (Fig. 9j). Interestingly this effect was inverted in the axons with rescuing expression of Syn2a^{WT}-mEos3.2 increasing the mobility of the super pool of SV, which was not achieved following re-expression of Syn2a^{K337Q}-mEos3.2 (Fig. 9k).'

In addition, the average diffusion co-efficient of synapsin2a in our study (Fig 9 J-K) aligns well with previous reports (Reshetniak *et al.*, 2020, <https://doi.org/10.15252/emj.2020104596>)

- The authors should discuss their results in light of the LLPS model. Can any links be drawn between both models, e.g. between multimerization and condensation? Can it not be interpreted in both ways or congruently?

We thank the reviewer for this question. We have amended our discussion to revolve around the potential links between synapsin LLPS and tetramerisation.

Discussion

“Synapsin isoforms have been shown to form oligomeric structures both in vitro and in vivo^{22,48,68}. Synapsins structural domain (C-domain) mediate its assembly into tetramers. Previous studies have identified key residues in the C-domain of synapsin that mediate its assembly into tetramers, that are essential for SV-tethering. One missense mutation known to perturb tetramer assembly, used in our study (K337Q) was unable to rescue reserve vesicle mobility and the SV density at the presynapse in SynTKO neuronal cultures. This suggests that homo-tetramerization of Syn2a is sufficient to rescue reserve pool mobility dynamics within the pre-synapse. Hetero-tetramerization, and/or synergistic/opposing roles of Synapsin1, in the background of Syn2a function, is likely to be important in regulating the SV cluster. It has been suggested that domain E (present in Synapsin1a, 2a and 3a isoforms) may adopt a different conformation, potentially during oligomerisation or within BMCs. In addition, such intramolecular interactions could regulate the targeting of synapsin isoforms into different presynaptic or axonal sub-cellular compartments.”

- What does AUC give us other than a qualitative measure of general mobility? Any quantitative interpretation?

AUC is used for the statistical comparisons of the MSD curves. The MSD provides quantitative assessment of the behaviour of each single molecule, and the average of this provides an assessment of overall confinement. We agree with the reviewer that more information of the trajectory dynamics can be added. Therefore, we have now added diffusion coefficient analysis to Figures: Fig. 3e, f; Fig. 4 and Fig. 9j, k. as mentioned above and requested by the reviewer.

The authors included some controls to check for biasing diffusion of unbound/free At565Nb. Maybe include more controls like the dye alone without Syt1pH?

As requested by the reviewer, we have performed an additional control experiment, now included in the results as Supplementary Fig 3.

Sup Fig.3

‘Supplementary Fig 3. Validation of anti-GFP ATTO-Nb specific binding to pHluorin (a) Representative image of a hippocampal neuron transfected with blue-fluorescent protein (cyan). (b) Max intensity projection of the same region in (a) after being stimulated for 5 minutes (high K^+ buffer containing anti-GFP Atto647-tagged nanobodies), the excess nanobodies washed off, and chased for 10 minutes to allow for internalisation of any bound nanobody (red). (c) Merged image. Scale bar 10 μm (a-c).’

Further, we have added this detail to the manuscript, mentioned in the Result section.

Results:

‘Further, to validate that the nanobody is selectively internalised into GFP-positive cargo, we applied anti-GFP-At647Nbs to hippocampal neurons transfected with cytosolic TagBFP, which is not extracellular facing, and we could not detect any internalised nanobodies or single molecules (Supplementary Fig 3).’

In these experiments, we expressed the blue fluorescent protein and exposed the neurons to the anti-GFP ATTO nanobodies. As expected, we saw a signal in the blue channel (405), but no nanobody was detected binding to any of the neurons (Far red channel; 647). These anti-GFP nanobodies have been generated by the company Synaptic Systems and are highly selective to GFP (Specificity recognizes: GFP (green fluorescent protein) and common GFP derivatives like EGFP, mEGFP, Sirius, tSapphire,

Cerulean, eCFP, mTurquoise, acGFP, Emerald, superecliptic pHluorin, paGFP, superfolder GFP, eYFP, mVenus and Citrine) and have been used in several publications from our lab (Joensuu *et al.*, 2016; Joensuu *et al.*, 2017; Gormal *et al.*, 2020, <https://doi.org/10.1073/pnas.2007443117>; Small *et al.*, 2021; doi: 10.1007/978-1-0716-1044-2_18; Joensuu *et al.*, 2020, doi: 10.1007/978-1-0716-0532-5_5).

- End page 8, beginning page 9: I got confused here. On one hand the authors state that the absence of synapsin did not dramatically alter the mobility patterns, on the other hand they state that synapsins selectively lower the mobility of the reserve pool. A slight rephrasing and indication of concrete numbers (diffusion coefficients) would help.

We thank the reviewer for this comment. We have rephrased the discussion section and included the diffusion coefficient data (see above).

- Figure 5: This quantification must include the labelling degree, otherwise it might be biased. Any comment/control? REPHRASE -

The reviewer is correct, we have now computed the ratio of recycling to reserve SVs for each condition. The ratio is by definition independent of the labelling efficiency which is low in both cases to enable single molecule imaging. We have rephrased to clarify this.

Results:

'When looking at the ratio of recycling to reserve SVs in WT and SynTKO neurons, we can see that there is a 3-fold increase in the number of recycling SVs in the SynTKO neurons indicating a potential shift in the sorting/segregation between the two populations of SVs (Fig. 6g, h). The altered ratio of reserve to recycling SVs in the SynTKO could reflect a loss of gatekeeping with the presynaptic boutons. It has been reported that approximately 80% of SVs are in the reserve pool in wild-type neurons². In agreement with this, we observed a higher proportion of reserve SVs to recycling SVs in wild-type neurons. In SynTKO neurons, the relative proportion of recycling to reserve SVs was lower. This is consistent with electron microscopy imaging showing a selective loss of reserve pool SVs in SynTKO presynaptic terminals¹. Our data agrees with the literature and suggests that synapsin functions as a gatekeeper which plays an active role in the transition/segregation between the two SV pools (Fig. 6e-h).'

- Page 10, middle: What stimulation? Name it.

We have now included the descriptor that the stimulation used was a high K⁺ buffer.

- Page 10, middle: What is meant by low MSD?

We have re-phrased this sentence as follows:

'Next, we examined the effect of high K⁺ stimulation on the mobility of synapsin2a expressed in SynTKO neurons, with Syt1pH co-expressed as a marker of presynaptic boutons. Analyses of the mobility of Syn2aWT-mEos3.2 showed an activity-dependent increase in its mobility as reflected by the MSD (Fig. 8a, b). In contrast, this effect was absent in the tetramerization-deficient synapsin2a mutant, which was already highly mobile (Fig. 8c, d).'

- Page 10, middle: What is meant by “regulates the nanoscale organisation”? Did not get the argument here.

The nanoscale organisation relates to how synapsin is dynamically organised at the synapse and axon using our nanoscopic measurements and how this organisation is affected by stimulation.

REVIEWERS' COMMENTS

Reviewer #1 (Remarks to the Author):

The authors have adequately addressed my comments from the previous round of review and I believe this paper has been substantially strengthened with both text and figure revisions.

I would like to endorse the acceptance of this paper for publication in NC.

Reviewer #2 (Remarks to the Author):

The authors have commented sufficiently to all my previous concerns and revised the manuscript very well accordingly. From that side I can only suggest publication.

However, in my opinion the authors have not well commented to the other referee's comment, especially on the conceptual novelty (first major comment of referee 1), which I think is a very valid point. This should be addressed more specifically.

Point-By-Point Response to the Reviewers:

Reviewer #1 (Remarks to the Author):

The authors have adequately addressed my comments from the previous round of review and I believe this paper has been substantially strengthened with both text and figure revisions.

I would like to endorse the acceptance of this paper for publication in NC.

We thank reviewer #1 for their kind words regarding our revision.

Reviewer #2 (Remarks to the Author):

The authors have commented sufficiently to all my previous concerns and revised the manuscript very well accordingly. From that side I can only suggest publication.

We thank reviewer #2 for their kind words regarding our revision.

However, in my opinion the authors have not well commented to the other referee's comment, especially on the conceptual novelty (first major comment of referee 1), which I think is a very valid point. This should be addressed more specifically.

This is an unusual request. Nevertheless, as clearly stated throughout the paper, the novelty of this manuscript lies in the demonstration that tetramerization of synapsin 2a controls the mobility (**Fig. 9f-i**) and presynaptic enrichment of the reserve SVs (**Fig. 9l**). Our work is the first super-resolution methodology capable of co-tracking SVs of the reserve and recycling vesicles in live hippocampal neurons, opening the way to many more studies into this intriguing SV clustering mechanism. This novel technique allows for direct observations of reserve SV mobility behaviour in and out of the presynapse and can be implemented to directly study the spatiotemporal interactions of reserve SVs with several presynaptic proteins in live imaging studies. We have demonstrated that synapsin 2a mobility and distribution are strongly affected by synaptic activity and these effects are dependent on the tetramerization of synapsin 2a (**Fig. 8**). Finally, we demonstrated, contrary to previous reports, that reserve pool vesicles are surprisingly found within axonal segments where they display similar mobility to that of recycling vesicles (**Fig. 6**). All these novelties have been extensively discussed in the manuscript.